# Semantic-Aware Motion Encoding for Topology-Agnostic Character Animation

**Zongye Zhang** [*] [1] [2]  **Yuzhuo Cui** [*] [1] [2]  **Qingjie Liu** [1] [2]  **Yunhong Wang** [1] [2]

## Abstract

Generalizing motion representation across diverse characters remains challenging due to significant topological variations in skeletal structures across datasets and species, which hinder the development of scalable generative models. To bridge this gap, we propose a Semantic-Aware Topology-Agnostic framework that learns a unified latent manifold shared by disparate species. Unlike methods relying on fixed hierarchies or rigid padding strategies, our approach leverages a semantic modulation mechanism to align functional joint correspondences, thereby decoupling motion from topology. This design enables the construction of a continuous, generative-friendly motion space from large-scale, unaligned raw BVH data. Experiments on human and animal datasets demonstrate that our framework achieves high-fidelity reconstruction and supports downstream text-to-motion tasks. Notably, the model enables zero-shot cross-species retargeting without paired data. Code and demos are available at https://github.com/zzysteve/SATA

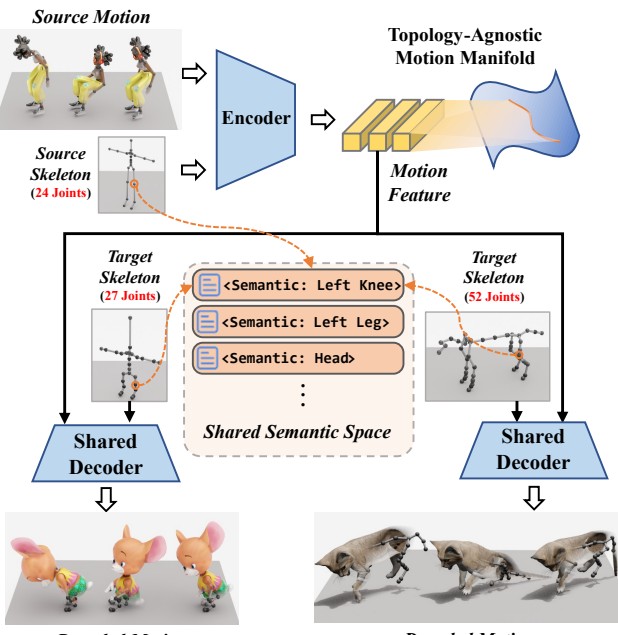

*Figure 1.* Zero-shot cross-species retargeting via a topology-agnostic motion manifold. Our method encodes source motion into a unified latent space, leveraging a shared semantic space to bridge disparate skeletal structures. The encoded features are then decoded into motion for diverse target topologies. This representation enables robust zero-shot generalization to heterogeneous skeletons, even without paired training data.

## 1. Introduction

Motion representation learning has long been a foundational field in computer graphics research, with extensive applications across various domains such as animation production, virtual reality, and robotics. Driven by advancements in multimodal generation, there is a growing demand for motion across diverse digital characters, from humans to fantastical creatures. This trend underscores the necessity of a unified framework capable of generalizing across biological structures. However, unlike the standardized structures of

images and audio, skeletal-based motion exhibits significant topological variations. Even within a single species like humans, joint definitions and hierarchies are often inconsistent across datasets (Plappert et al., 2016; Mahmood et al., 2019; Lin et al., 2023b), posing a substantial barrier to unified representation learning.

Existing topology-flexible approaches either rely on local graph connectivity (Lee et al., 2023) or padded joint sequences (Gat et al., 2025; Lee et al., 2025), leaving functional correspondences ambiguous and introducing redundant computation under heterogeneous skeletons. This motivates a compact semantic-aware encoder that aligns diverse topologies without paired cross-species supervision.

Constrained by these variations, most existing motion generation methods (Tevet et al., 2023; Guo et al., 2024; Wang

---

[*]Co-first authors. Zongye Zhang led the project and is listed first among co-first authors.  [1]State Key Laboratory of Virtual Reality Technology and Systems, Beihang University, Beijing, China [2]Hangzhou Innovation Institute, Beihang University, Hangzhou, Zhejiang, China. Correspondence to: Qingjie Liu <qingjie.liu@buaa.edu.cn>.

*Proceedings of the 43[rd] International Conference on Machine Learning*, Seoul, South Korea. PMLR 306, 2026. Copyright 2026 by the author(s).

et al., 2025; Zhang et al., 2025c; 2026b;a) rely on domain-specific skeletal representations, restricting their applicability to specific topologies. While recent graph-based approaches like SAME (Lee et al., 2023) offer topological flexibility, they rely on local structural connectivity and lack the explicit semantic grounding required for cross-species alignment. Conversely, padding-based Transformers (Gat et al., 2025; Lee et al., 2025) achieve any-topology generation instead of a compact representation learning. Additionally, these methods impose strict constraints on joint counts and rely on rigid zero-padding, which hinder the learning of a continuous and structured latent manifold. Consequently, a critical gap remains for an efficient, semantics-aware encoder that intrinsically adapts to arbitrary topologies.

To bridge this gap, we propose a Semantic-Aware Topology-Agnostic (SATA) motion autoencoder, as illustrated in Figure 1. Our key insight is that despite topological diversity, motion maintains inherent semantic consistency across biological entities. To capture this synergy, we propose Semantic-aware Feature Modulation, which injects automatically augmented semantic joint tags along with their spatial positions. To strengthen dynamic modeling, we incorporate a Spatio-Temporal Interleaved architecture designed to capture long-range motion evolution. By integrating semantic joint alignment with temporal trajectory modeling, our method effectively bridges heterogeneous topologies and complex dynamics within a unified latent space for robust motion representation. Concretely, SATA takes a motion sequence with an arbitrary BVH hierarchy as a dynamic graph, encodes it into a topology-independent latent, and decodes it to a target skeleton conditioned only on rest-pose geometry and semantic tags. It therefore does not require a canonical joint set, source-target motion pairs, or hand-crafted joint correspondences.

To fully leverage the scaling potential of our topology-agnostic encoder, we broaden the scope of existing benchmarks by introducing a topology-adaptive data processing pipeline. This pipeline ensures direct compatibility with industry-standard raw BVH formats, enabling the seamless integration of diverse motion resources in their native structures. It also facilitates large-scale joint training across diverse datasets and species, paving the way for scalable biological motion representation learning.

We validate the effectiveness of our framework on human (Guo et al., 2022) and animal (Wang et al., 2025) datasets processed via our proposed pipeline. Experimental results demonstrate that our framework effectively disentangles abstract motion semantics from specific skeletal topologies, achieving high-fidelity motion reconstruction and retargeting. Notably, our model exhibits zero-shot cross-species retargeting capability, prioritizing the preservation of source motion intentionality even when mapping between disparate locomotion modes (e.g., bipedal to quadrupedal). These results highlight the robustness and generative potential of our learned universal motion manifold.

Our contributions can be summarized as follows:

- **Unified Generative Motion Representations.** We present a novel topology-agnostic autoencoder designed to learn a scalable and transferable motion representation. By decoupling motion semantics from topological constraints, our model captures intricate dynamics and serves as a robust foundation for both retargeting and generative tasks.

- **Semantic-Aware and Spatio-Temporal Designs.** We introduce two core components: a Semantic-Aware Feature Modulation that aligns functional correspondences across diverse species, and a Spatio-Temporal Interleaved architecture designed to capture intricate dynamics without topological constraints.

- **Scalability and Zero-Shot Generalization.** Our topology-agnostic design overcomes data scarcity barriers by supporting large-scale joint training across heterogeneous datasets. Extensive experiments demonstrate not only superior reconstruction performance, but also **zero-shot cross-species retargeting** capabilities.

## 2. Related Works

### 2.1. Skeletal Motion Representation

Human motions represented in joint rotations (Loper et al., 2015; Guo et al., 2022) are high-dimensional spatial-temporal signals, which are often difficult for neural networks to model. To facilitate large-scale training, recent large-scale benchmarks (Plappert et al., 2016; Guo et al., 2022; Wang et al., 2025) are built on the assumption of a canonical skeleton template, which requires rigorous data normalization. Raw motion sequences in SMPL (Loper et al., 2015) format are mapped to a specified topology with a fixed number of joints and hierarchy. Building upon these canonical formats, various frameworks utilize VAE (Petrovich et al., 2021; Athanasiou et al., 2022; Lin et al., 2023a; Zhang et al., 2025d) or VQ/RVQ-VAE (Zou et al., 2024; Guo et al., 2024; Pinyoanuntapong et al., 2024; Zhang et al., 2025a;b) architectures to map high-dimensional joint trajectories onto compact latent representations. Although such approaches ensure high-fidelity reconstruction, they are fundamentally constrained by a fixed structural prior, which hinders their applicability to heterogeneous skeletal topologies.

Recent retargeting methods use skeleton-aware pooling or residual correction (Aberman et al., 2020; Zhang et al., 2023;

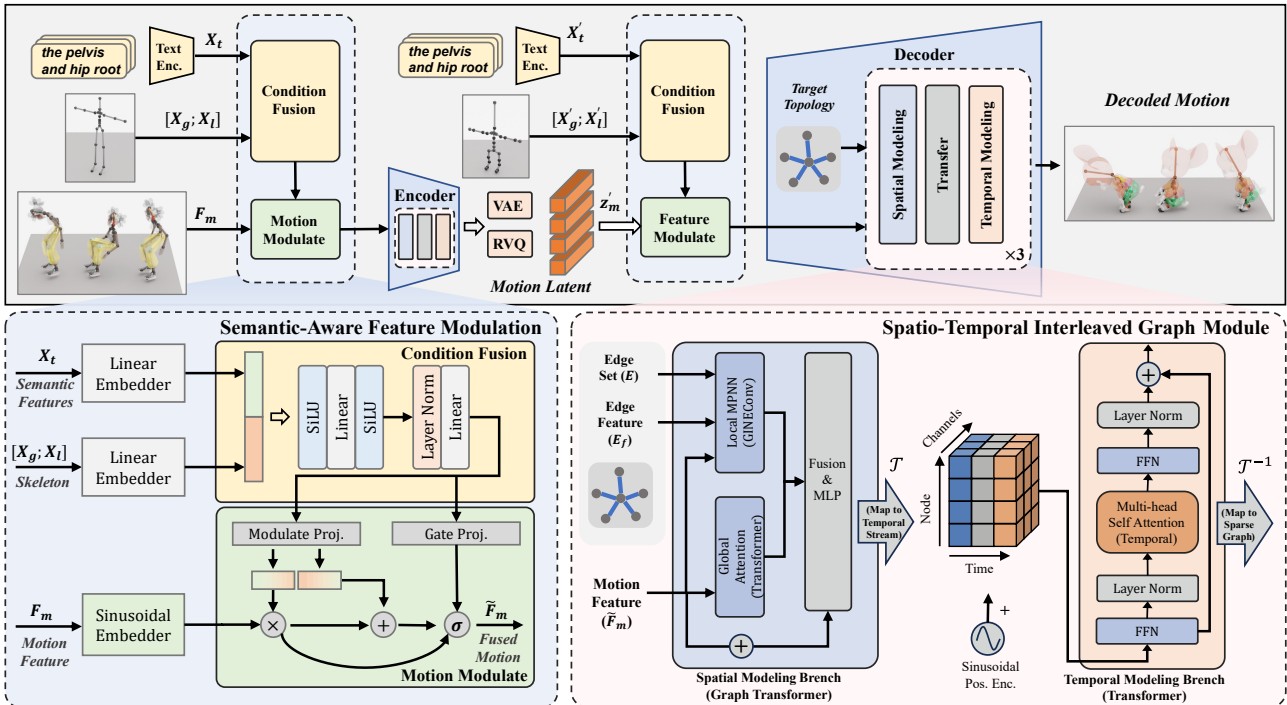

*Figure 2.* Overview of the proposed motion autoencoder. Our framework utilizes Semantic-aware Feature Modulation to condition motion features $F_m$ with source semantics $X_t$ and skeletal structure $\{X_g, X_l\}$. The core pipeline consists of symmetrical encoder-decoder layers based on Spatio-Temporal Interleaved Graph Blocks. The latent space $z_m$ is regularized via VAE or RVQ-VAE and combined with target structural priors $\{X'_t, X'_g, X'_l\}$ to perform topology-agnostic motion reconstruction.

2024c) and graph autoencoding (Lee et al., 2023), but they mostly remain tied to human or prepared character rigs.

## 2.2. Heterogeneous and Any-Topology Modeling

While GCNs (Yan et al., 2018) naturally model skeletal graphs, they are primarily optimized for discriminative tasks like action recognition. SAME (Lee et al., 2023) extends to graph-based autoencoders, offering a robust framework for handling topological variations. However, given its design focus on isomorphic structures, establishing consistent functional correspondences across heterogeneous species can be ambiguous without explicit semantic grounding.

Conversely, recent generative approaches (Gat et al., 2025; Lee et al., 2025) employ Transformers with zero-padding to handle diverse skeletons. However, these methods are restricted to implicit learning for synthesis and lack the capability for explicit motion encoding, which limits their potential as versatile backbones for downstream applications. Furthermore, the padding strategy incurs quadratic computational redundancy proportional to the maximum padded capacity and imposes constraints on joint counts. Unlike these generative approaches, our work constructs a padding-free autoencoder that intrinsically adapts to arbitrary topologies without such structural constraints.

## 2.3. Cross-Species Motion Modeling

Related cross-structure animation systems use canonical motion spaces or shape-aware mesh handles (Zhang et al., 2024a;b), yet they target stylization or skeleton-free synthesis rather than reusable topology-agnostic motion encoding.

Existing approaches generally bridge the topological gap through either explicit mapping mechanisms or template-constrained latent alignment. Some works rely on manual intervention: (Egan et al., 2024) employs rule-based retargeters for initialization, while Motion2Motion (Chen et al., 2025) demands explicit keypoint correspondences, fundamentally restricting their applicability to scenarios with pre-defined mappings. Conversely, latent-based methods attempt to bypass pairings but remain bound by rigid structural constraints. WalkTheDog (Li et al., 2024) requires training character-specific autoencoders for each morphology, limiting scalability. While (Zhang et al., 2025c) explores unpaired cross-species transfer, it remains confined to fixed quadrupedal templates. Similarly, OmniMotionGPT (Yang et al., 2024) is tied to fixed SMAL (Zuffi et al., 2017) skeletal templates. However, these dependencies on target-specific training or rigid templates prevent true generalization to arbitrary, variable topologies. In contrast, our approach enables **topology-agnostic** modeling without explicit pairings,

utilizing a *semantic-aware module* to capture shared motion semantics across disparate biological entities.

## 3. Methods

### 3.1. Motion Representation

Unlike previous methods that rely on sampling a fixed set of joints (Guo et al., 2022; Wang et al., 2025), which often discards structural details, we preserve the complete skeletal hierarchy by formulating motion as a sequence of graphs. To facilitate joint training across heterogeneous species, we introduce a unified topology-agnostic graph representation that can be directly generated from raw BVH files.

Formally, a motion sequence with $T$ frames is represented as an ordered set of graphs $\mathcal{G} = \{G_1, G_2, \ldots, G_T\}$. For each frame $t$, the skeleton is parameterized as an undirected graph $G_t = (V, E, E_f, F_s, F_{m,t})$, where $V$ and $E$ define the set of joints and their connectivity, respectively. To encapsulate the structural hierarchy, we introduce structural edge features $E_f$, which are defined by the topological depth and reverse depth of the connected child joint.

The node features are further decomposed into static and dynamic components. The static skeleton features $F_s = (X_g, X_l, X_t)$ capture time-invariant properties: $X_g \in \mathbb{R}^3$ denotes the global offset w.r.t. root, $X_l \in \mathbb{R}^3$ represents the rest-pose coordinate relative to the parent, and $X_t$ signifies the encoded joint semantic features which provide the necessary grounding for cross-species alignment. The dynamic motion features $F_{m,t}$ characterize the temporal evolution of the pose. Specifically, for a given frame with $J = |V|$ joints, we define $F_m = (q, x, v_q, v_x, r, c) \in \mathbb{R}^{J \times D}$, where $D$ is the per-joint feature dimension. Here, $q$ utilizes a 6D rotation representation (Zhou et al., 2019) and $x$ denotes the relative position of joints to the parent. To model motion dynamics, $v_q$ and $v_x$ represent the rotational and linear velocities, respectively. Following (Guo et al., 2022), the root motion $r = (\dot{r}^a, \dot{r}^x, \dot{r}^z, r^h)$ captures the angular velocity around the Y-axis, horizontal linear velocities, and root height. $c$ is automatically derived from joint height and speed to enforce physical grounding.

### 3.2. Semantic-Aware Feature Modulation

Traditional skeletal modeling primarily relies on spatial coordinates and hierarchical depths. However, these purely geometric features fail to provide the semantic information to establish functional correspondences across disparate biological topologies. To address these limitations, we leverage Multimodal Large Language Models (MLLMs) to generate enriched semantic descriptions for each joint. In our implementation, Gemini 2.5 Pro generates the descriptions and a frozen T5 encoder (Raffel et al., 2020) embeds them into $X_t$. We employ a standardization strategy that enforces

taxonomic neutrality and structural generalization. This ensures consistent functional grounding across species and facilitates scalable expansion, as new datasets can be aligned by referencing established description protocols. Representative descriptions and robustness checks are provided in Appendix Section C.4.

To deeply integrate semantic embeddings, we move beyond simple addition and propose a Semantic-Spatial Modulation mechanism inspired by Feature-wise Linear Modulation (Perez et al., 2018). This approach allows the model to dynamically reshape motion features based on the unique semantic-spatial context of each joint, as illustrated in Figure 2.

Specifically, the model first embeds three heterogeneous inputs: motion features $\mathbf{F}_m$, spatial coordinates $\mathbf{X}_g, \mathbf{X}_l$, and semantic embeddings $\mathbf{X}_t$.

$$\mathbf{z}_m = \phi_m(\mathbf{F}_m), \ \mathbf{z}_s = \phi_s([\mathbf{X}_g; \mathbf{X}_l]), \ \mathbf{z}_t = \phi_t(\mathbf{X}_t) \quad (1)$$

where $\phi_m$ and $\phi_t$ are linear projection layers, $[\cdot; \cdot]$ denotes the concatenation operation. Notably, $\phi_s$ denotes a Sinusoidal Encoder that maps continuous spatial coordinates into high-frequency spectral embeddings, enabling the model to perceive fine-grained spatial relationships.

To capture the contextual identity of each node, the spatial embeddings and semantic features are concatenated and processed through a non-linear mapping network $\Phi$ to generate a joint modulation condition $\mathbf{c} = \Phi([\mathbf{z}_s; \mathbf{z}_t])$. The condition vector $\mathbf{c}$ encapsulates the functional essence of a node, which is then used to predict a pair of modulation parameters: the scaling factor $\gamma$ and the shifting factor $\beta$ via a modulation projection $[\gamma, \beta] = \Psi(\mathbf{c})$.

The motion features $\mathbf{z}_m$ are then normalized and actively reshaped by these parameters:

$$\hat{\mathbf{x}} = \text{LN}(\mathbf{z}_m) \odot (1 + \gamma) + \beta \quad (2)$$

where $\text{LN}(\cdot)$ represents Layer Normalization and $\odot$ denotes the Hadamard product.

To ensure adaptive integration and mitigate potential noise from conflicting modalities, we introduce a gating mechanism $g$ to control the information flow. The final fused representation $\widetilde{\mathbf{F}}_m$ is obtained via a gated residual connection:

$$\widetilde{\mathbf{F}}_m = \mathbf{z}_m + g \odot \hat{\mathbf{x}}, \quad \text{where} \quad g = \sigma(\mathbf{W}_g \mathbf{c}) \quad (3)$$

Here, $\sigma$ is the Sigmoid activation function and $\mathbf{W}_g$ represents learnable weights. This design ensures that the model can selectively emphasize or suppress semantic-spatial information based on the specific topological context.

### 3.3. Spatio-Temporal Interleaved Graph VAE

Traditional motion models (Guo et al., 2024) often flatten joints into fixed-size vectors, which discards important structural details. In contrast, we represent motions as dynamic graphs to maintain topological adaptability. The core challenge lies in balancing intra-frame biophysical constraints (e.g., bone lengths, joint limits) with inter-frame kinetic coherence. To this end, we propose a Spatio-Temporal Interleaved Graph architecture that iteratively synchronizes spatial and temporal modeling. By alternating between spatial graph convolutions to enforce physical bone constraints and temporal attention to guide logical evolution, this interleaved design prevents common artifacts such as joint jittering or structural drifting.

**Spatial Modeling Branch.** To capture complex bio-constraints across heterogeneous skeletons, each spatial block employs a dual-path spatial module inspired by GP-SConv (Rampášek et al., 2022). We utilize GINEConv (Hu et al., 2020) as a message-passing engine to enforce local structural priors, which introduces relative hierarchical depths to enhance the awareness of skeletal topology. Parallel to this, a Spatial Transformer models long-range synergies, such as the coordinated movements between disparate limbs that are not directly connected in the graph. Finally, the outputs from both paths are aggregated via summation and refined by an MLP, followed by a residual connection to ensure robust feature integration.

**Temporal Modeling Branch.** The temporal branch focuses on motion fluidity through long-term dependency modeling. Inspired by TimeSformer (Bertasius et al., 2021), we implement a Spatio-Temporal Interleaving mechanism. For each joint, we extract its features across the temporal dimension to establish long-term dependencies. We define a mapping operator $\mathcal{T}$ that reshapes the batched graph sequence $\mathcal{G}_{batch}$ into continuous joint-wise temporal streams $\mathcal{X}_{temp} = \mathcal{T}(\mathcal{G}_{batch})$. A Temporal Transformer, equipped with sinusoidal positional encodings, operates on these streams to capture high-order dynamics. The updated stream $\mathcal{X}'_{temp}$ is then redistributed back into the graph structure via an inverse mapping $\mathcal{G}_{updated} = \mathcal{T}^{-1}(\mathcal{X}'_{temp})$.

**Unified Latent Representation.** Our symmetric framework employs a spatial max pooling layer to aggregate node-wise features into a frame-level latent vector $\mathbf{z}_t$, intentionally stripping away source-specific structural constraints to form a topology-independent manifold. This latent space is regularized using either VAE (Kingma & Welling, 2014) or RVQ-VAE (van den Oord et al., 2017; Guo et al., 2024) mechanisms: the VAE variant is used for the main reconstruction and retargeting experiments, while RVQ-VAE provides discrete tokens for token-based generation. During decoding, this latent vector is broadcast to all nodes of the target skeleton, and a final MLP predicts $\hat{F}_m^{out} = (q, r, c) \in \mathbb{R}^{J \times 11}$ for

motion recovery via forward kinematics.

### 3.4. Training Objectives and Inference

Our model is trained end-to-end by optimizing a composite objective $\mathcal{L} = \mathcal{L}_{rec} + \lambda \mathcal{L}_{reg}$. For reconstruction $\mathcal{L}_{rec}$, we follow the methodology of SAME (Lee et al., 2023), combining reconstruction MSE (on rotations, positions, and velocities) with physical regularization for foot contact, ground penetration, and smoothness. These constraints ensure the decoded motion remains physically valid across heterogeneous skeletons. For latent regularization $\mathcal{L}_{reg}$, we adapt the objective to the chosen bottleneck: a KL-divergence loss is applied for the continuous VAE setup to enforce a Gaussian prior, while a commitment loss is utilized for the discrete RVQ-VAE framework to align the encoded latents with the learned codebook. During inference, we use overlapping sliding windows and crop the overlapping frames from the latter window.

## 4. Experiments

### 4.1. Implementation Details

#### 4.1.1. MODELS

The encoder and decoder share a symmetric layout with three Spatio-Temporal Interleaved blocks. Unless otherwise specified, motion, skeleton, and text features are fused into a hidden width of 256. Each block applies graph message passing over the skeletal structure followed by temporal self-attention; the Transformer components use four attention heads, dropout 0.1, and a feed-forward inner width of 512. Edge messages are updated by a two-layer ReLU MLP. For the VAE, a small linear layer maps the 128-D latent code to the mean and log-variance of a diagonal Gaussian; for RVQ-VAE, we use 256-D latents with 6 residual quantizers and a codebook size of 1024.

To ensure capacity is not the main source of improvement, we widen the SAME baseline for fair comparison. Our base model has 8.41M parameters and 29.66 GFLOPs, while SAME has 5.98M parameters and 20.21 GFLOPs. The modest increase is mainly due to the Transformer branch in our graph blocks. We train all models with Adam for 400 epochs using an initial learning rate of 0.0001, a 30-epoch linear warmup, and exponential decay with $\gamma = 0.99$ and a minimum factor of 0.01. During generation, we use 64-frame windows with a 16-frame overlap for smooth transitions.

#### 4.1.2. DATASET

As existing large-scale benchmarks (Plappert et al., 2016; Guo et al., 2022; Lin et al., 2023b; Wang et al., 2025) are predominantly constructed upon standardized skeletal tem-

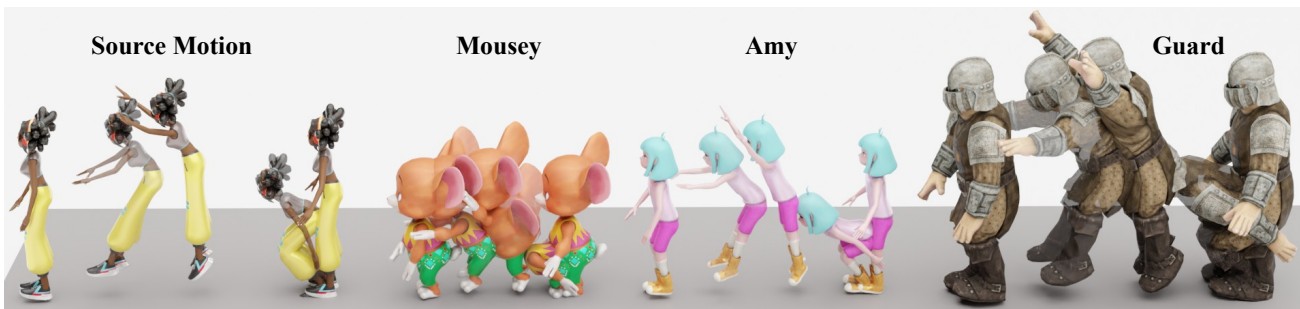

*Figure 3.* **Qualitative retargeting results on the AT-HumanML3D dataset.** Given a single source motion (left), our model encodes it and then decodes it to multiple target characters with distinct body proportions and skeletal scales (Mousey, Amy, and Guard).

*Table 1.* **Single-Dataset Training Evaluation.** Comparisons of reconstruction quality when trained on single datasets. Gray cells denote zero-shot cross-dataset evaluation. "-" indicates topology-dependent methods inapplicable to disparate skeletons.

| METHOD | TEST ON AT-HUMANML3D | | | | | TEST ON AT-ANIMO4D | | | | |
|---|---|---|---|---|---|---|---|---|---|---|
| | JR↓ | RT↓ | JP↓ | FS↓ | GP↓ | JR↓ | RT↓ | JP↓ | FS↓ | GP↓ |
| *Training Source: AT-HumanML3D* | | | | | | | | | | |
| MOMASK | 0.8087 | 14.066 | 25.389 | 0.0087 | 0.0462 | - | - | - | - | - |
| SAME | 0.0831 | 2.1635 | 2.8102 | **0.0004** | 0.0006 | 0.5721 | 191.46 | 398.14 | 0.2851 | 0.1803 |
| OURS | **0.0568** | **0.9583** | **1.3570** | 0.0006 | **0.0005** | 0.4855 | 15.593 | 34.585 | 0.3516 | 1.5545 |
| *Training Source: AT-AniMo4D* | | | | | | | | | | |
| MOMASK | - | - | - | - | - | 0.4137 | 15.699 | 27.651 | 0.0708 | 0.1948 |
| ANIMO | - | - | - | - | - | 0.4354 | 17.765 | 30.913 | 0.0145 | 0.0331 |
| SAME | 0.5266 | 110.51 | 122.69 | 0.6441 | 3.5076 | 0.5227 | **1.4390** | **4.3032** | 0.0088 | **0.0025** |
| OURS | 0.6616 | 57.228 | 80.908 | 0.0085 | 0.0412 | **0.3901** | 1.6515 | 4.5063 | **0.0087** | 0.0055 |

plates, we employ a dedicated data preparation pipeline to extract arbitrary-topology motion data from the HumanML3D (Guo et al., 2022) and AniMo4D (Wang et al., 2025) datasets for validation.

We constructed two variable-topology benchmarks by converting the original motion data into quaternion-based BVH formats. To ensure robustness across diverse morphologies, we applied data augmentation to the human subset following (Lee et al., 2023) to introduce varied skeletal proportions. All sequences underwent canonicalization for initial states and facing directions, followed by the computation of topology-agnostic representations as defined in Section 3.1. The resulting any-topology benchmarks include AT-HumanML3D, which comprises 80,508 sequences with a total duration of 724.6 minutes, and AT-AniMo4D, which contains 30,097 sequences covering 115 animal species with a total duration of 539.3 minutes. Comprehensive processing details and statistics are provided in Appendix Section C.

### 4.1.3. EVALUATION METRICS

We evaluate the reconstructed motions via two protocols: universal geometric metrics (Lee et al., 2023) to assess structural fidelity across disparate topologies, and task-specific semantic metrics (Guo et al., 2022) for human benchmarks.

**Geometric Metrics.** To assess the structural precision of reconstructed motions, we employ a suite of geometric metrics. These include joint rotations (**JR**) for local postural accuracy, and global positions of the root (**RT**) and joints (**JP**) to evaluate character displacement and trajectory consistency. Furthermore, to verify physical plausibility, we report foot skating (**FS**) and ground penetration (**GP**) artifacts, which quantify the realism of the interaction between the character and the ground. Please refer to Appendix for more details.

**Semantic Metrics.** To evaluate high-level semantic alignment, we leverage our model's inherent retargeting capability to interface with the original HumanML3D benchmark. We apply a transformation to project our topology-agnostic representation back into standardized HumanML3D coordinate space. We then compute established metrics, including Top-3 Retrieval Accuracy and Multi-Modality Distance (**MMD**), to measure motion-text consistency. Additionally, Fréchet Inception Distance (**FID**) is reported to quantify the distributional discrepancy between our reconstructed features and the ground-truth motion manifold.

### 4.2. Reconstruction Evaluation

**Single-Dataset Training Evaluation.** Table 1 reports the reconstruction performance. For comparison, we adapted two topology-dependent methods by introducing a zero-padding

*Table 2.* **Multi-Dataset Joint Training Evaluation.** Comparison of topology-agnostic encoders when trained jointly on AT-HumanML3D and AT-AniMo4D.

| METHOD | TEST ON AT-HUMANML3D | | | | | TEST ON AT-ANIMO4D | | | | |
|---|---|---|---|---|---|---|---|---|---|---|
| | JR↓ | RT↓ | JP↓ | FS↓ | GP↓ | JR↓ | RT↓ | JP↓ | FS↓ | GP↓ |
| SAME | 0.1060 | 1.5534 | 2.3416 | **0.0004** | **0.0004** | 0.2357 | 1.4112 | 3.6803 | **0.0079** | **0.0021** |
| OURS | **0.0769** | **1.1426** | **1.6311** | 0.0005 | **0.0004** | **0.1971** | **1.2340** | **3.2744** | 0.0091 | 0.0064 |

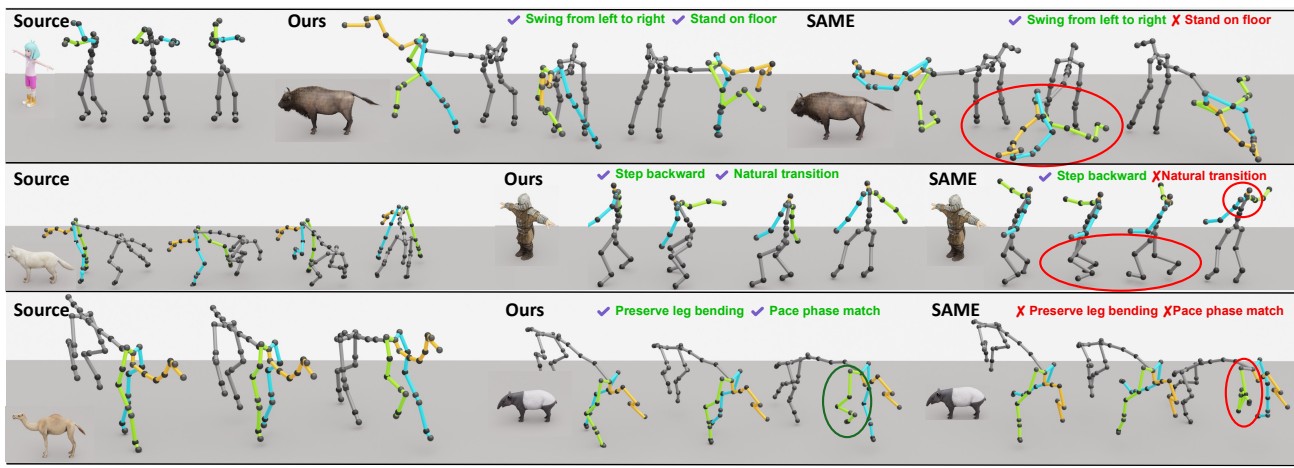

*Figure 4.* **Zero-shot retargeting comparison.** The figure illustrates transfers across diverse species, generated by the VAE model jointly trained on both datasets. Our method outperforms baseline (Lee et al., 2023) by maintaining structural stability and preserving nuanced motion semantics, whereas the baseline exhibits distortion and motion loss in cross-topology scenarios. We recommend viewing the demo video for a clearer comparison of dynamic quality.

*Table 3.* **Retargeting Capability on AT-HumanML3D.**

| METHOD | INTERNAL | CROSS |
|---|---|---|
| MOMASK | 89.421 | 103.72 |
| SAN | 15.9647 | 34.8207 |
| SAME | 1.4842 | 0.9604 |
| OURS (RVQ) | 1.1238 | 0.9669 |
| OURS (VAE) | **0.2122** | **0.1974** |

strategy. The implementation details are provided in Appendix Section D.3. Note that topology-dependent methods are omitted in cross-settings as they cannot naturally adapt to disparate skeletal structures. Regarding cross-dataset transfer results highlighted in gray, our method demonstrates robust zero-shot generalization. The gray cross-dataset cells therefore evaluate OOD generalization across training sources, with target skeletons drawn from an unseen dataset domain. Notably, in the Human→Animal setting, SAME suffers from catastrophic failure with JP around 398, whereas our model maintains integrity with JP around 35. Note that lower FS/GP scores do not necessarily indicate better realism in globally failed cases. On the AT-AniMo benchmark, although SAME achieves lower trajectory errors, our method yields superior pose fidelity. Crucially, as

shown in the joint training results Table 2, introducing human motion data effectively enhances our trajectory modeling, allowing our unified framework to outperform baseline methods comprehensively.

**Multi-Dataset Joint Training.** Table 2 demonstrates the superior scalability of our framework in handling heterogeneous data. In the challenging joint training setting, our method consistently outperforms the baseline (Lee et al., 2023), achieving approximately **30%** lower reconstruction errors (JR, RT, JP) on AT-HumanML3D and **12%** lower on the diverse AT-AniMo4D benchmark. Although SAME maintains competitive scores on local physical metrics (FS and GP), the absolute gaps are small and it struggles with global motion coherence. In contrast, our approach prioritizes global structural fidelity and semantic consistency, demonstrating superior generalization as discussed in Section 5.1.

### 4.3. Motion Retargeting

Adhering to the protocol in (Lee et al., 2023), we evaluate retargeting on AT-HumanML3D using global joint position error (see Appendix Section D.2 for details). As shown in Table 3, our VAE model demonstrates superior retarget-

*Table 4.* **Comprehensive component ablation on AT-HumanML3D.** We report reconstruction and retargeting errors under component removals; w/o S. W. disables the sliding window strategy while keeping all components.

| METHOD | COMPONENT | | | | | RECONSTRUCTION | | | | | RETARGETING | |
|---|---|---|---|---|---|---|---|---|---|---|---|---|
| | GNN | S-TF | T-TF | FUS. | TEXT | JR[RAD]↓ | RT[CM]↓ | JP[CM]↓ | FS↓ | GP[CM]↓ | INT.↓ | CROSS↓ |
| OURS (BASE) | ✓ | ✓ | ✓ | ✓ | ✓ | 0.0568 | 0.9583 | 1.3570 | 0.0006 | 0.0005 | 0.2122 | 0.1974 |
| W/O S. W. | ✓ | ✓ | ✓ | ✓ | ✓ | 0.0630 | 3.1599 | 3.4472 | 0.0010 | 0.0004 | 0.7548 | 0.6485 |
| *Spatial Modeling* | | | | | | | | | | | | |
| W/O SPATIAL TF | ✓ | ✗ | ✓ | ✓ | ✓ | 0.1243 | 8.8678 | 10.228 | 0.0006 | 0.0001 | 1.7797 | 1.6705 |
| W/O GNN | ✗ | ✓ | ✓ | ✓ | ✓ | 0.0747 | 1.5616 | 2.3750 | 0.0031 | 0.0004 | 1.0978 | 1.0610 |
| SPATIAL ONLY | ✓ | ✓ | ✗ | ✗ | ✗ | 0.0648 | 1.0841 | 1.6612 | 0.0008 | 0.0017 | 1.0367 | 0.9910 |
| *Temporal Modeling* | | | | | | | | | | | | |
| W/O TEMPORAL TF | ✓ | ✓ | ✗ | ✓ | ✓ | 0.0584 | 1.3726 | 1.7776 | 0.0005 | 0.0005 | 0.3209 | 0.2858 |
| *Semantic Modulation* | | | | | | | | | | | | |
| W/O FUSION BLOCK | ✓ | ✓ | ✓ | ✗ | ✗ | 0.1196 | 6.9959 | 7.8066 | 0.0020 | 0.0004 | 0.4802 | 0.4776 |
| W/O TEXT FUSION | ✓ | ✓ | ✓ | ✓ | ✗ | 0.0663 | 1.0936 | 1.5245 | 0.0005 | 0.0004 | 0.3525 | 0.2470 |

*Table 5.* **Effectiveness of Pre-training.** We compare reconstruction errors of models trained from scratch (✗) vs. fine-tuned from AT-HumanML3D (✓) across different data scales.

| DATA | PT | RECONSTRUCTION METRICS | | | | |
|---|---|---|---|---|---|---|
| | | JR ↓ | RT ↓ | JP ↓ | FS ↓ | GP ↓ |
| 10% | ✗ | 0.6201 | 5.2580 | 14.3442 | 0.0579 | 0.1769 |
| | ✓ | **0.2825** | **3.5179** | **7.7118** | **0.0458** | **0.0858** |
| 30% | ✗ | 0.5244 | 3.3640 | 8.7174 | **0.0160** | **0.0149** |
| | ✓ | **0.2205** | **1.3612** | **4.0628** | 0.0190 | 0.0217 |
| 100% | ✗ | 0.3901 | 1.6515 | 4.5063 | **0.0087** | **0.0055** |
| | ✓ | **0.1822** | **1.1530** | **3.1403** | 0.0116 | 0.0082 |

*Table 6.* **Text-to-motion generation on AT-HumanML3D.**

| METHOD | SEMANTIC | | | REALISTIC | |
|---|---|---|---|---|---|
| | FID↓ | MMD↓ | TOP-3↑ | FS↓ | GP↓ |
| *Upper Bound (Ground Truth)* | | | | | |
| GT | 0.000 | 2.901 | 0.808 | 0.0433 | 0.1040 |
| *Text-to-Motion Generation* | | | | | |
| SAME | 1.628 | 4.661 | 0.554 | 0.0091 | 0.0023 |
| OURS (VAE) | 1.226 | 4.576 | 0.563 | 0.0063 | **0.0014** |
| OURS (RVQ) | **0.158** | **3.955** | **0.639** | **0.0015** | 0.0017 |

in Appendix Section B.

ing fidelity, outperforming SAN (Aberman et al., 2020) and SAME by a large margin. To investigate alternative encoding paradigms, we also adapted MoMask as a proxy for padding-based approaches. The significant performance gap indicates that while padding works well for fixed topologies, directly applying it to retargeting scenarios faces inherent challenges in alignment.

## 4.4. Ablation Studies

We validate our design choices in Table 4. The spatial modeling rows show that both graph message passing and spatial attention are important for structural fidelity, with removing the *Spatial Transformer* causing degradation. The *Sliding Window* strategy mainly affects global trajectory consistency, as reflected by the sharp increase in translation error without it. The *Fusion Block* and *Text Fusion* rows confirm that semantic-spatial modulation is critical for topology-aware alignment, while the temporal branch further improves trajectory-level coherence. These results support our design philosophy: semantic fusion aligns heterogeneous joints, spatial modeling preserves skeletal structure, temporal modeling stabilizes motion dynamics, and the full model combines these roles for robust reconstruction and retargeting. Additional inference ablations can be found

# 5. Additional Analysis

In this section, we explore the zero-shot behavior and extended applications of our topology-agnostic encoder, demonstrating its generalization capability across disparate biological structures and its potential as a foundation for downstream tasks.

## 5.1. Zero-shot Cross-species Retargeting

Remarkably, our framework demonstrates zero-shot cross-species retargeting without exposure to any paired human-animal data. Due to the lack of paired human-animal datasets with rich semantics for quantitative benchmarking, we further validate this through qualitative comparisons in Figure 4 and videos on our project page. These results indicate that our semantic-aware modulation effectively establishes functional correspondences between disparate skeletons, such as implicitly aligning human lower limbs with quadrupedal rear legs to preserve locomotion semantics.

## 5.2. Cross-Species Transfer Learning

We investigate the data efficiency of our framework by leveraging the model pre-trained on AT-HumanML3D as a

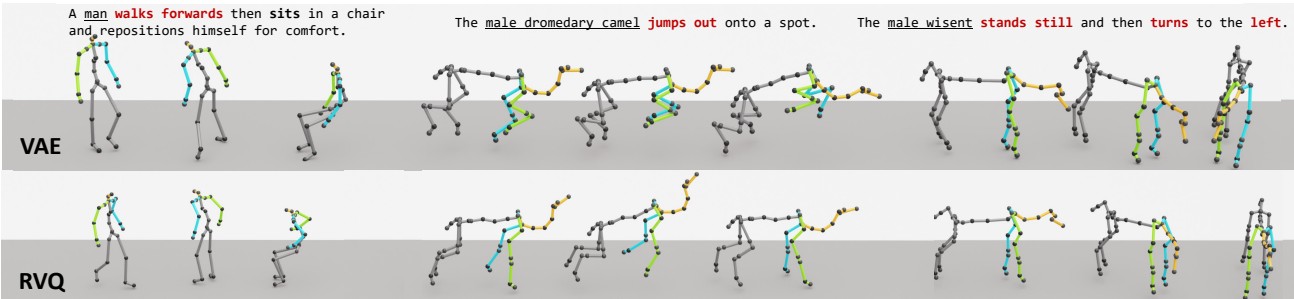

*Figure 5.* **Visualization of text-to-motion generation**. We show generated motions for human and animal subjects conditioned on textual descriptions. The top and bottom rows display results generated from VAE and RVQ versions. Both variants produce high-quality motions that accurately reflect the textual semantics (highlighted in red).

foundation. We finetune this backbone on varying subsets of the AT-AniMo4D dataset. As shown in Table 5, pre-training yields consistent performance gains over training from scratch. Notably, in the low-data regime (10%), the pre-trained model reduces the Joint Rotation (JR) error by over **50%**. This confirms that our encoder captures transferable motion priors from human data, enabling effective transfer to diverse animal morphologies even with sparse data.

### 5.3. Downstream Task: Text-to-Motion

To further validate the quality and generative potential of our learned manifold, we extend our framework to the text-to-motion synthesis task. We employ representative diffusion-based (Tevet et al., 2023) and token-based (Guo et al., 2024) backbones to generate motion representations from textual descriptions. Concretely, MDM predicts continuous VAE latents, while MoMask predicts discrete RVQ codes; both are decoded by our motion decoder. Quantitative results are presented in Table 6, with visualization in Figure 5. The results demonstrate the effective learnability and structural regularity of our latent representations. Although there remains room for improvement compared to models optimized exclusively for human data, our approach prioritizes broad generalization across biological structures. This establishes a promising baseline for topology-agnostic generation.

## 6. Conclusion

In this paper, we introduce a Semantic-Aware Topology-Agnostic framework designed to learn a unified generative manifold from large-scale, heterogeneous data. By effectively decoupling motion semantics from structural constraints, our approach overcomes the fragmentation caused by skeletal variations, enabling high-fidelity reconstruction and generative modeling across diverse biological topologies. Extensive experiments validate that this unified rep-

resentation not only supports robust downstream tasks but also enables zero-shot cross-species retargeting. While our current implementation demonstrates the scalability of joint training, future work will focus on further expanding data diversity and model capacity to solidify this foundation for broad character animation.

**Discussion.** Our current scope is topology-agnostic representation for diverse biological articulated systems, rather than arbitrary mechanical structures or all possible rare anatomies. Zero-shot retargeting may still produce physically implausible details when the source dynamics exceed the target body's feasible motion range. Future work will formalize quantitative protocols for cross-species transfer, scale data and model capacity toward rare skeletons, and incorporate physics-aware or anatomy-aware priors for more executable motions.

## Impact Statement

Our work advances the field of machine learning by introducing a topology-agnostic representation for heterogeneous motion data. The potential societal consequences are largely positive, including the acceleration of digital character animation, virtual reality, and interactive experiences. However, as with any generative technology, there is a possibility of misuse in creating hyper-realistic synthetic content. Furthermore, although we utilize a large-scale dataset, we acknowledge the importance of data diversity to avoid bias in representing rare species. We encourage the responsible use of these motion priors in both creative and scientific contexts.

## Acknowledgements

This work is supported by "Pioneer" and "Leading Goose" R&D Program of Zhejiang (No. 2024C01020).

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

# A. More Qualitative Results

## A.1. Zero-shot Cross-species Retargeting

The project page provides several demonstration videos illustrating our framework's capacity for zero-shot cross-species retargeting. We evaluate these capabilities using AT-AniMo and AT-HumanML3D as the primary sources for animal and human motion data, respectively. Three distinct transfer scenarios are explored: (i) human-to-animal, (ii) animal-to-human, and (iii) animal-to-animal. Notably, none of these cross-species pairings were encountered during the training phase. The model needs to autonomously establish semantic correspondences across disparate skeletal configurations, reconstructing plausible motions directly from topology-agnostic latent embeddings.

The accompanying videos demonstrate that our framework effectively captures the global motion dynamics of the source character and the coordinated movements of corresponding anatomical segments. These features are subsequently remapped onto the target skeleton during the decoding phase. This capability persists even across non-isomorphic skeletal structures (e.g., human-to-quadruped transitions), where the model preserves the semantic essence of the source motion while accommodating the target's unique morphology.

**Discussion.** We acknowledge that the zero-shot nature of this task introduces inherent constraints. In the absence of explicit pair-wise supervision, synthesized poses for certain target species may occasionally appear biomechanically implausible or unnatural for their specific anatomy. Such artifacts typically emerge when the source motion dynamics significantly exceed the anatomical range of motion or the inherent structural constraints of the target skeleton. To address these challenges, future research will explore the transition from implicit latent modeling to explicit semantic construction and physics-aware constraints. By harmonizing unified cross-species modeling with structured anatomical priors, it is possible to ensure that retargeted motions are not only semantically consistent but also physically viable and executable across the vast diversity of biological morphologies.

## A.2. Text-to-Motion Generation

On the project page, we provide a comprehensive set of video demonstrations for Text-to-Motion (T2M) generation, covering both human subjects and diverse animal species.

For human motion, the visualizations confirm that our topology-agnostic manifold preserves the visual fidelity and semantic consistency required for high-quality generation. The synthesized motions exhibit realistic kinematics and align well with complex textual descriptions, demonstrating the effectiveness of our encoding strategy on standard topologies.

Transitioning to the cross-species domain (AT-AniMo), the generation task becomes significantly more challenging due to the extreme structural diversity and the scarcity of high-quality motion data for specific species. Despite these obstacles, our model successfully learns a structured and coherent motion space that supports the synthesis of semantically recognizable animal behaviors. While achieving perfectly naturalistic motion for arbitrary creatures remains an open challenge, our framework establishes a promising foundation for topology-agnostic generative modeling.

## A.3. In-domain Motion Retargeting

On our project page, we provide additional video demonstrations focusing on in-domain motion retargeting, specifically targeting human-to-human transfers across diverse body shapes and proportions.

**Morphological Adaptation.** Although our primary focus is cross-species generalization, these results highlight the model's high-fidelity performance in handling intra-species morphological variations. By leveraging the AT-HumanML3D dataset, which encompasses a diverse range of human skeletal scales and bone lengths, we demonstrate that our framework effectively decouples core motion dynamics from specific body proportions. As demonstrated in the videos, the same source motion is successfully adapted to target skeletons ranging from tall and slender characters to shorter and more robust figures.

**Adaptation of Motion Scale and Dynamics.** A key observation in these in-domain results is the model's ability to adapt the scale of motion according to the target's physical dimensions. For instance, in jumping sequences, the model automatically adjusts the global displacement and jump distance to match the target's limb lengths and overall skeletal scale.

### A.4. Motion Reconstruction

To validate the encoding capability of our framework, we evaluate motion reconstruction, where ground-truth motions are first encoded into the latent space and subsequently decoded back onto the identical source skeleton for comparison. We provide video visualizations of these results on both the AT-HumanML3D and AT-AniMo datasets.

**Analysis of Human Motion.** On the AT-HumanML3D dataset, both the VAE and RVQ-VAE variants achieve near-perfect reconstruction. The reconstructed motions are structurally stable and kinematically indistinguishable from the ground truth, demonstrating the model's effective compression of standard human dynamics.

**Analysis of Animal Motion.** On the more challenging AT-AniMo benchmark, the VAE variant exhibits slightly superior reconstruction fidelity compared to the discrete RVQ-VAE. While the model successfully preserves the global semantic content and locomotion patterns, we observe minor deviations in local high-frequency details compared to the human setting. This performance gap is attributed to the inherent complexity of the animal dataset, which encompasses extreme structural diversity and more erratic dynamic variations.

## B. More Ablation Studies

### B.1. Impact of Sliding Window Overlap

Since our model is trained on fixed-length motion clips (e.g., 64 frames), direct inference on longer sequences often results in temporal discontinuities and trajectory drift at clip boundaries. To mitigate this, we employ a sliding window strategy with blending during inference.

As presented in Table 7, applying the sliding window strategy significantly enhances reconstruction fidelity across both bottleneck architectures (VAE and RVQ). Without sliding windows ("✗"), the model suffers from severe global drift, evidenced by high Root Trajectory (RT) and Joint Position (JP) errors. For instance, in the RVQ setting, the RT error drops dramatically from 13.33 to 3.59 after introducing the sliding window mechanism. This confirms that the overlapping and blending operations are crucial for preserving the global temporal consistency of long motion sequences.

*Table 7.* **Ablation on sliding window strategy.** We compare the reconstruction performance with and without the sliding window strategy across different encoder bottlenecks.

| BOTTLENECK | SLIDING WINDOW | OVERLAP SIZE | RECONSTRUCTION | | | | |
|---|---|---|---|---|---|---|---|
| | | | JR↓ | RT↓ | JP↓ | FS↓ | GP↓ |
| VAE | ✗ | N/A | 0.0630 | 3.1599 | 3.4472 | 0.0010 | **0.0004** |
| | ✓ | 0 | 0.0600 | 1.1643 | 1.6014 | **0.0005** | 0.0005 |
| | ✓ | 16 | **0.0568** | **0.9583** | **1.3570** | 0.0006 | 0.0005 |
| RVQ | ✗ | N/A | 0.2034 | 13.3341 | 16.4800 | 0.0226 | 0.0021 |
| | ✓ | 0 | **0.1669** | 3.6193 | 5.0194 | 0.0009 | **0.0005** |
| | ✓ | 16 | 0.1670 | **3.5855** | **4.9822** | **0.0008** | **0.0005** |

### B.2. Impact of Sliding Window Size

We further investigate the sensitivity of the model to different sliding window configurations, specifically varying the window size and overlap stride. As shown in Table 8, the reconstruction quality is positively correlated with the window size.

The configuration with a larger window size consistently outperforms smaller settings. This is attributed to the fact that human and animal locomotion typically involves temporal dependencies that span several seconds. A small window restricts the encoder's temporal receptive field, preventing it from capturing complete motion cycles, which leads to increased errors in global trajectory (RT) and root stability. Consequently, we adopt a window size of 64 with a 16-frame overlap as our default setting to balance computational efficiency and temporal coherence.

*Table 8.* **Ablation on sliding window configurations.** The metrics evaluate the reconstruction quality under different window sizes and overlap strides using VAE as bottleneck. Larger windows provide better temporal context.

| SLIDING WINDOW | RECONSTRUCTION | | | | |
|---|---|---|---|---|---|
| (SIZE/OVERLAP) | JR↓ | RT↓ | JP↓ | FS↓ | GP↓ |
| 64 / 16 | **0.0568** | **0.9583** | **1.3570** | 0.0006 | 0.0005 |
| 32 / 8 | 0.0627 | 2.2030 | 2.7490 | **0.0005** | 0.0003 |
| 16 / 4 | 0.0667 | 3.2405 | 3.7970 | 0.0027 | **0.0002** |

## C. More Details on Dataset

### C.1. Any-Topology HumanML3D

The original HumanML3D dataset is provided in SMPL format, and we utilize the CAT tool (Fukahire, 2025) to convert the axis-angle based SMPL representations into quaternion-based BVH format. Subsequently, all BVH files undergo a canonicalization process, where the initial root position is translated to the origin, and the coordinate system is aligned such that $Y+$ is the up-axis with the character facing the $Z+$ direction.

Since the original HumanML3D employs mirror augmentation, we implement a corresponding augmentation on the BVH files to ensure alignment with existing evaluation protocols. Once the BVH files are obtained, we transform the motions into the representation described in Section 3.1. This representation allows for lossless conversion back to BVH format, enabling our model's outputs to be directly utilized in standard animation pipelines without the need for complex post-processing or additional retargeting operations.

To facilitate robust modeling across diverse human body proportions, we perform skeletal augmentation using Autodesk MotionBuilder (Autodesk, 2025), following the methodology in (Lee et al., 2023). To ensure a fair comparison with previous methods, we utilize the same set of characters, where each motion sequence is paired with two distinct target skeletons. Consequently, the resulting AT-HumanML3D dataset contains 80,508 sequences, totaling 724.6 minutes of motion data.

Regarding the dataset split, we follow the original HumanML3D partitions for training, validation, and testing. The training set consists of 64,242 samples. For reconstruction and generation tasks, the validation and test sets contain 1,338 and 4,042 samples, respectively. For retargeting evaluation, the validation set comprises 4,014 samples, while the test set contains 12,126 samples.

### C.2. Any-Topology AniMo4D

*Table 9.* Overall statistics of the AT-AniMo4D dataset across training, validation, and test splits.

| Split | # Species | # Samples | Ratio (%) | Male | Female | Juvenile | # Actions |
|---|---|---|---|---|---|---|---|
| Train | 89 | 23,275 | 77.33 | 8,526 | 6,306 | 8,443 | 535 |
| Val | 12 | 3,351 | 11.13 | 1,183 | 1,104 | 1,064 | 280 |
| Test | 14 | 3,471 | 11.53 | 1,371 | 832 | 1,268 | 286 |
| **Total** | **115** | **30,097** | **100.0** | **11,080** | **8,242** | **10,775** | **1,101** |

For the AniMo4D (Wang et al., 2025) dataset, we implement a rigorous data cleaning pipeline following the acquisition of raw BVH files, including deduplication and the removal of corrupted motion sequences. Similar to our processing of HumanML3D, we apply a canonicalization process where the initial root position is translated to the origin, and the coordinate system is aligned such that $Y+$ is the up-axis with the character facing the $Z+$ direction. Since AniMo4D does not provide paired ground-truth for retargeting, quantitative evaluations on this dataset are restricted to reconstruction tasks. The resulting AT-AniMo4D dataset comprises 30,097 sequences across 115 animal species, totaling 539.3 minutes, with macroscopic statistics summarized in Table 9.

**Phylogenetic and Morphological Splitting.** To rigorously evaluate the model's topology-agnostic generalization, we moved beyond the original AniMo4D categories and proposed a custom split based on taxonomic families and skeletal archetypes. This ensures that the model is tested on its ability to infer motion for "unseen" yet biologically related structures. The rationale for our split is as follows:

*Table 10.* Detailed species statistics for the validation and test sets in AT-AniMo4D.

| Split | Species Name | Samples | Male | Female | Juvenile | Actions |
|-------|--------------|---------|------|--------|----------|---------|
| **Val** | Clouded Leopard | 377 | 74 | 150 | 153 | 185 |
| | Siberian Tiger | 375 | 126 | 131 | 118 | 168 |
| | Blue Wildebeest | 299 | 101 | 97 | 101 | 109 |
| | Baird's Tapir | 287 | 98 | 98 | 91 | 103 |
| | Red Deer | 284 | 99 | 88 | 97 | 112 |
| | Spotted Hyena | 273 | 90 | 97 | 86 | 114 |
| | American Alligator | 256 | 89 | 92 | 75 | 104 |
| | Red Panda | 248 | 139 | 0 | 109 | 155 |
| | Gray Wolf | 243 | 94 | 96 | 53 | 104 |
| | Dingo | 241 | 88 | 72 | 81 | 110 |
| | Thomson's Gazelle | 235 | 94 | 87 | 54 | 98 |
| | Japanese Macaque | 233 | 91 | 96 | 46 | 113 |
| **Test** | Snow Leopard | 403 | 151 | 136 | 116 | 175 |
| | Red Ruffed Lemur | 356 | 120 | 124 | 112 | 144 |
| | Sand Cat | 314 | 186 | 0 | 128 | 188 |
| | Dama Gazelle | 302 | 104 | 102 | 96 | 106 |
| | Arctic Wolf | 295 | 104 | 105 | 86 | 117 |
| | Dromedary Camel | 282 | 106 | 101 | 75 | 113 |
| | Okapi | 248 | 86 | 91 | 71 | 99 |
| | Wisent | 214 | 116 | 0 | 98 | 119 |
| | Spectacled Caiman | 212 | 69 | 69 | 74 | 97 |
| | Asian Small-clawed Otter | 202 | 94 | 0 | 108 | 112 |
| | Red-necked Wallaby | 182 | 50 | 45 | 87 | 103 |
| | Somali Wild Ass | 180 | 98 | 0 | 82 | 99 |
| | Fennec Fox | 161 | 49 | 59 | 53 | 70 |
| | Malayan Tapir | 120 | 38 | 0 | 82 | 92 |

**Training Set (88 Species):** The training set is designed to maximize the coverage of the global motion manifold. It spans a diverse array of skeletal hierarchies, including heavy-set ungulates (e.g., Elephants, Rhinos), agile felids and canids, primates, reptiles (e.g., Crocodiles), and unique outliers (e.g., Anteaters, Pangolins). This diversity encourages the model to learn a broadly transferable representation capable of handling disparate limb-to-trunk ratios and connectivity patterns.

**Validation Set (12 Species):** We selected 12 representative species with high-quality samples (typically $> 200$ sequences) and balanced gender distributions to serve as stable monitors for convergence. These include archetypal members from major families, such as the Siberian Tiger (Felids), Gray Wolf (Canids), and Blue Wildebeest (Ungulates), as well as structural outliers like the Baird's Tapir (unique spine structure) and Spotted Hyena (sloping hindquarters).

**Test Set (14 Species):** The test set is specifically curated to evaluate zero-shot generalization. We selected species that were excluded from the training set but possess "morphological cousins" within it (e.g., Snow Leopard vs. training Jaguar, Arctic Wolf vs. training Gray Wolf). This setup challenges the model to generalize its learned semantic priors to novel topologies that share functional similarities with training data but differ in specific skeletal proportions.

The detailed species distribution and individual sample statistics for the validation and test sets are provided in Table 10.

### C.3. Semantic Tag Augmentation

The effectiveness of our Semantic-Aware Modulation (SeAM) depends on the quality of the joint-level semantic priors. To bridge the gap between raw, inconsistent joint names in various datasets and a unified functional manifold, we develop a Semantic Augmentation Pipeline.

Our pipeline analyzes existing datasets like HumanML3D and AniMo4D. While these datasets provide skeletal hierarchies, their joint naming conventions are often abbreviated (e.g., "L_UpLeg" in SMPL vs. "Front_Left_Thigh" in animal skeletons). We leverage Multimodal Large Language Models (MLLMs) as autonomous technical artists to parse these raw strings. By providing the MLLM with both the joint name and its parent-child context (e.g., Joint [Parent]), the pipeline expands sparse labels into detailed functional descriptors, as illustrated in Figure 6. For instance, a simple "frontLegUpr_joint" label is augmented to "The upper bone of the front leg". This process ensures that disparate biological structures share a canonical

semantic grounding, allowing the model to recognize functional homologies across humans and animals. To further enhance the model's resilience to varying chain lengths, our pipeline implements numerical invariance. By instructing the MLLM to ignore numeric indices (e.g., mapping both "Spine_1" and "Spine_2" to a generalized "Spine Segment" descriptor), we ensure that the semantic embeddings $X_t$ remain consistent across species with different skeletal resolutions.

To handle arbitrary skeletons with non-standard naming or unknown hierarchies, our framework offers a plug-and-play interface that requires only minimal user intervention. Instead of the tedious manual bone-remapping or full-skeleton rigging required by traditional methods, users only need to provide sparse semantic tags for a few key joints (e.g., labeling the "anchor" joints like hips, knees, or paws).

Once these key semantic anchors are provided, our framework utilizes the same augmentation pipeline to project these sparse inputs into the Universal Semantic Space. The SeAM module then propagates this semantic guidance across the graph structure, effectively attaching the novel topology to our learned motion manifold. This design significantly reduces the technical barrier for cross-species retargeting, allowing users to animate any custom skeleton by simply defining its functional essence through a handful of natural language tags.

*Table 11.* **Representative semantic descriptions for AT-HumanML3D.** We show a compact subset of the generated dictionaries; the full joint-description mappings for all training skeletons are provided with the code release.

| Raw Joint | Original | Variant A | Variant B |
|---|---|---|---|
| Hips | the pelvis and hip root anchor | Pelvis | Hips |
| LowerBack | a lower back bone segment | Lower Back | Lower Back |
| Spine1 | a spine bone segment | Lower Thoracic Spine | Spine 1 |
| Head_End | the head crown tip | Head Crown | Head End |
| LeftShoulder | the left shoulder clavicle bone | Left Clavicle | Left Shoulder |
| LeftArm | the left upper arm bone | Left Upper Arm | Left Arm |
| LeftLeg | the left lower leg calf bone | Left Calf | Left Leg |
| LeftToeBase_End | the left toes tip | Left Toe Tip | Left Toe Base End |

### C.4. Robustness of Semantic Tag Generation

**Description Variants.** Since semantic descriptions are encoded as part of the training input, changing them requires regenerating semantic tags and retraining the model rather than simply swapping text at inference time. We therefore evaluate representative description variants on a 10% subset of AT-HumanML3D under the same architecture and training protocol, as listed in Table 11. As shown in Table 12, replacing verbose descriptions with shorter alternatives such as "Pelvis" or "Hips" produces only modest performance changes, indicating robustness to reasonable paraphrases.

**MLLM Variants.** To evaluate sensitivity to the choice of language model, we regenerated semantic tags with different MLLMs under the same prompt constraints and trained on a 10% subset of AT-HumanML3D, while keeping the architecture and evaluation protocol unchanged. As shown in Table 13, downstream reconstruction and retargeting performance remain comparable across MLLM variants, indicating that our constrained vocabulary and prompt design reduce semantic-label variance.

*Table 12.* **Robustness to semantic description variants.** We regenerate the semantic tags and retrain on a 10% subset of AT-HumanML3D for each variant.

| TEMPLATE | RECONSTRUCTION | | | | | RETARGETING | |
|---|---|---|---|---|---|---|---|
| | JR[RAD]↓ | RT[CM]↓ | JP[CM]↓ | FS↓ | GP[CM]↓ | INTERNAL↓ | CROSS↓ |
| ORIGINAL | **0.1555** | **1.8504** | **3.0299** | **0.0004** | 0.0004 | 1.5744 | **1.4198** |
| VARIANT A | 0.1636 | 2.5392 | 3.7029 | 0.0010 | **0.0003** | 1.6766 | 1.5326 |
| VARIANT B | 0.1573 | 2.2694 | 3.3667 | **0.0004** | 0.0003 | **1.5708** | 1.4288 |

*Table 13.* **Robustness to different MLLM-generated semantic tags.** We report reconstruction and retargeting performance on AT-HumanML3D.

| MLLM | RECONSTRUCTION | | | | | RETARGETING | |
|---|---|---|---|---|---|---|---|
| | JR[RAD]↓ | RT[CM]↓ | JP[CM]↓ | FS↓ | GP[CM]↓ | INTERNAL↓ | CROSS↓ |
| GEMINI 2.5 PRO | **0.1555** | **1.8504** | **3.0299** | **0.0004** | **0.0004** | **1.5744** | **1.4198** |
| LLAMA 3.3 70B | 0.1559 | 2.0458 | 3.1362 | 0.0013 | 0.0005 | 1.7486 | 1.5479 |
| QWEN3-NEXT 80B | 0.1666 | 2.2954 | 3.3013 | 0.0007 | 0.0005 | 1.7207 | 1.6201 |
| GPT-5.4-PRO | 0.1652 | 1.9429 | 3.0642 | 0.0011 | 0.0005 | 1.5782 | 1.4661 |

## D. More Implementation Details

### D.1. Semantic Metrics and Evaluation Protocol

To evaluate high-level semantic alignment using standard benchmarks, we leverage our model's zero-shot retargeting capability to interface with the HumanML3D evaluator. Specifically, we project the generated topology-agnostic representations onto the canonical SMPL skeleton (using HumanML3D sample ID `000021` as the target template). The retargeted motions are then converted into global world coordinates and subsequently processed into the standard 263-dimensional HumanML3D feature space using the official processing pipeline.

Based on these projected features, we compute standard metrics including Top-3 Retrieval Accuracy (R-Precision), Multi-Modality Distance (MMD), and Fréchet Inception Distance (FID). It is important to note that this evaluation protocol introduces an inherent structural challenge for our method. Unlike baseline models trained directly on the fixed HumanML3D topology, our model's outputs undergo a two-stage transformation: (1) zero-shot retargeting to the SMPL topology and (2) feature re-extraction. This process inevitably induces a distributional shift and minor geometric deviations compared to the ground truth manifold. Consequently, metrics sensitive to exact geometric alignment (e.g., R-Precision) may reflect this conversion overhead rather than a lack of semantic consistency, whereas distributional metrics like FID remain robust indicators of generation quality.

### D.2. Retargeting Metrics

Following the evaluation protocols in SAME (Lee et al., 2023), we categorize the retargeting tasks into two settings: *intra-dataset retargeting*, where the source and target share identical skeletal definitions, and *cross-dataset retargeting*, which involves transfer between diverse human skeletal structures.

To accurately quantify the retargeting fidelity while accounting for scale variations, we adopt the character-normalized Mean Squared Error (MSE) as proposed in SAN (Aberman et al., 2020). Specifically, the error is calculated based on the global positions of the joints, normalized by the square of the target character's height. The metric is formally defined as:

$$E_{\text{retarget}} = 1000 \times \frac{1}{T \cdot J} \sum_{t=1}^{T} \sum_{j=1}^{J} \frac{|\mathbf{p}t, j - \hat{\mathbf{p}}t, j|^2}{H^2} \tag{4}$$

where $\mathbf{p}_{t,j}$ and $\hat{\mathbf{p}}_{t,j}$ denote the global positions of the $j$-th joint at frame $t$ for the ground truth and the generated motion, respectively. $H$ represents the height of the target character, used as a normalization factor to ensure scale invariance. $T$ and $J$ refer to the total number of frames and joints. The final value is scaled by a factor of 1000 for readability.

### D.3. Baseline Implementation Details

Due to the scarcity of encoder frameworks explicitly designed for variable-topology motion, direct baselines for our task are unavailable. To establish rigorous benchmarks, we adapted two representative approaches: MoMask (Guo et al., 2024), representing the state-of-the-art in fixed-topology human motion generation, and AniMo (Wang et al., 2025), representing advanced cross-species motion modeling. We introduced a unified padding strategy to bridge the topological gap, enabling these topology-dependent methods to process heterogeneous skeletons.

### D.3.1. ADAPTATION OF MOMASK TO ARBITRARY TOPOLOGIES

MoMask originally operates on a fixed set of joints with a pre-defined order. To adapt this fixed-topology RVQ-VAE to arbitrary topologies with varying numbers of joints and connectivity, we transform the frame-wise skeletal pose graph into a **fixed-length vector** via zero-padding. This allows the utilization of the original 1D CNN backbone without architectural modification.

Specifically, for each frame, we extract the root feature $r \in \mathbb{R}^4$ and joint features $x_j \in \mathbb{R}^{d_j}$, which comprise local and global positions, rotations, and velocities. Given a dataset with a maximum joint count $J_{\max}$, for any skeleton with $J \leq J_{\max}$ actual joints, we stack the joint features and zero-pad the sequence to match $J_{\max}$:

$$\tilde{X} = [x_1, \ldots, x_J, \underbrace{\mathbf{0}, \ldots, \mathbf{0}}_{J_{\max}-J}] \in \mathbb{R}^{J_{\max} \times d_j}. \tag{5}$$

The padded joint features are then flattened and concatenated with the root feature to form a fixed-dimension input vector $z$:

$$z = [r; \mathrm{vec}(\tilde{X})] \in \mathbb{R}^{4+J_{\max} \cdot d_j}. \tag{6}$$

Consequently, the input tensor for the sequence becomes $Z \in \mathbb{R}^{B \times T \times (4+J_{\max}d_j)}$. To handle the variable valid lengths, we construct a binary padding mask $M \in \{0,1\}^{B \times T \times J_{\max}}$, where entries corresponding to padded joints are marked as invalid. This mask is applied during loss calculation and metric evaluation to exclude zero-padded regions. It is important to note that the MoMask backbone **does not explicitly model the skeletal topology**, such as kinematic chains, in this adaptation; topological variations are implicitly handled via alignment in the padded vector space.

### D.3.2. ADAPTATION OF ANIMO TO ARBITRARY TOPOLOGIES

We reproduced AniMo and adapted it to our variable-topology setting. While AniMo was originally designed with joint-level spatial modeling, applying it to our heterogeneous dataset required specific data alignment strategies to handle varying joint counts and connectivity within a unified batch.

**Unified Input and Masking.** To make the disparate skeletons from our dataset compatible with AniMo's batch processing, we employed the same zero-padding strategy used for MoMask. We aligned all skeletal data to a fixed maximum joint count $J_{\max}$ by padding the joint feature sequence with zeros. A binary mask $M$ was generated to distinguish between valid joints and padded regions, ensuring that the model could be trained on our mixed-species data without architectural conflict.

**Structured Tokenization with Contact Tokens.** Consistent with the original AniMo design, we decomposed the input features into global root data, non-contact joint features, and contact information. To accommodate the varying number of contact points across different species, we aggregated the contact features into a single **contact pseudo-joint token**. Consequently, the input sequence for the spatial transformer was constructed as:

$$\mathcal{T} = \{t_{\mathrm{root}}\} \cup \{t_{\mathrm{joint}}^{(1)}, \ldots, t_{\mathrm{joint}}^{(J_{\max})}\} \cup \{t_{\mathrm{contact}}\}. \tag{7}$$

This formulation allows the contact distribution to interact with somatic joints via the attention mechanism, maintaining the structural integrity of the original method.

**Masked Spatial Attention.** During the spatial encoding phase, we utilized the generated padding mask $M$ within the self-attention layers of the Spatial Transformer. By setting the attention weights of padded tokens to negative infinity, we ensured that the spatial modeling focused exclusively on valid joints for each specific topology. This allowed the original AniMo backbone to effectively learn spatial correlations from our variable-topology data without being disrupted by the zero-padded artifacts.

**Text-Conditioned Adaptation.** To leverage AniMo's capability for semantic control, we extracted species and gender descriptions from our dataset metadata. These text descriptions were encoded using a pre-trained CLIP model and injected into the temporal convolutional network via Feature-wise Linear Modulation, or FiLM. This setup provided the necessary semantic guidance, helping the baseline model distinguish between the diverse morphological structures present in our cross-species training set.

### D.4. Text-to-Motion Implementation Details

To evaluate the generative capabilities within our learned topology-agnostic manifold, we conducted text-to-motion experiments utilizing text-motion pairs from the original HumanML3D and AniMo4D datasets. Based on the bottleneck structure of our motion encoder, we categorized the pre-encoded motion sequences into two types: **discrete codes** derived from the RVQ-VAE bottleneck and **continuous latent representations** derived from the VAE bottleneck. We adapted prominent generative frameworks to model these representations, respectively.

**Discrete Representation (MoMask).** For the discrete setting, we adopted the Masked Transformer architecture from MoMask (Guo et al., 2024). While the original MoMask models discrete codes derived from a fixed-topology RVQ-VAE (specifically trained on HumanML3D), we adapted the transformer to predict the topology-agnostic discrete code indices produced by our unified RVQ encoder. The training objective follows the original masked modeling paradigm, where the model learns to reconstruct masked code indices conditioned on CLIP-encoded text descriptions. During inference, the model iteratively predicts the discrete codes to form the motion sequence.

**Continuous Representation (MDM).** For the continuous setting, we employed the Motion Diffusion Model (MDM) (Tevet et al., 2023) as the generative backbone. In this configuration, the diffusion process operates on the **continuous latent vectors** $z$ extracted from our VAE encoder. This effectively functions as a Latent Diffusion Model, where the model is trained to denoise Gaussian noise into coherent motion latent representations conditioned on text prompts.

**Inference and Evaluation.** Upon generation, the predicted latent representations are fed into our pretrained, topology-dependent decoder to synthesize the final motion sequences with the specified target topology. These decoded motions are then subjected to quantitative and qualitative evaluations to assess the semantic consistency and generation quality.

# Role
You are a Senior Technical Artist and Biomechanics Specialist specializing in Cross-Species Motion Retargeting. Your goal is to map raw 3D joint names from heterogeneous skeletons (Humans, Quadrupeds, etc.) into **Universal Semantic Labels** that serve as grounding for Text-to-Motion models (CLIP/T5).

# Context
Our framework uses these labels to build a "Topology-Agnostic" latent space. We need labels that describe the **functional role** of the joint, ensuring that a "Human Knee" and a "Dog Stifle" share the same semantic grounding.

# Task
Analyze the provided "Joint List" (often with parent-child hierarchy hints) and generate a JSON mapping: {"raw_name": "semantic_description"}.

# Rules for Universal Semantic Grounding
1. **Taxonomic Neutrality**:
   - Use terms that apply to both bipeds and quadrupeds where possible.
   - "UpLeg" -> "Upper Hind Leg / Thigh".
   - "Arm" -> "Front Limb / Arm".
   - "Hand/Foot" -> "Terminal End of Front/Hind Limb".
2. **Hierarchy-Aware Interpretation**:
   - If the raw name is ambiguous (e.g., "Leg"), look at its position:
     - Root -> Spine -> Leg (means Thigh/Upper Leg).
     - Thigh -> Leg (means Calf/Shin).
3. **Functional Role Expansion**:
   - Don't just clean the name; describe its locomotion role.
   - "Hips/Pelvis" -> "Root / Pelvis Area (Center of Motion)".
   - "ToeBase" -> "Toe Base (Support Pivot)".
4. **Handle End Sites & Effectors**:
   - Joints ending in "_End", "Nub", or "Site" signify extremities.
   - Map to "... Tip / Effector" (e.g., "Head Top Tip", "Front Left Paw Tip").
5. **Standardize Orientation**:
   - Always expand: "L/l" -> "Left", "R/r" -> "Right", "Fr" -> "Front", "Bk" -> "Back".
6. **Numerical Invariance (Aggregation)** :
   - Ignore numeric suffixes or indices that denote sequential segments of the same body part (e.g., "Spine1", "Spine_2", "Neck_3").
   - Map these to the same core functional description (e.g., all map to "Spine segment").
# Mode: {{ MODE }}

{% if MODE == "ALIGNMENT" %}
# Alignment Constraints (Cross-Species Dictionary)
You must ensure these labels align with our **Universal Motion Vocabulary**.
**Requirement**:
1. Search the [Existing Vocabulary] below. If a joint plays the same functional role as an entry there, use that entry's exact string.
2. If the structure is unique to this species (e.g., "Tail", "Wing"), create a new clean label following the rules above.

[Existing Reference Vocabulary] :
{{ REFERENCE_VOCAB_LIST }}
{% endif %}

# Input Data (Format: Raw Joint Name [Parent Name])
{{ RAW_JOINTS_LIST }}

# Output Format
Output ONLY a valid JSON object.
Example: {"root_joint": "The center of gravity of the body", "Spine2": "A spine bone segment"}

JSON:

*Figure 6.* Prompt design for cross-species joint semantic alignment. The prompt template enforces rules such as taxonomic neutrality and numerical invariance to guide the MLLM in extracting functional homologies from heterogeneous skeletons.

