# OpenReview forum: "Semantic-Aware Motion Encoding for Topology-Agnostic Character Animation"
_ICML.cc/2026/Conference — ICML 2026 regular_

### Official Review · Reviewer_hWbJ · 2026-03-03

**Soundness:** 2
**Presentation:** 2
**Significance:** 3
**Originality:** 3
**Overall Recommendation:** 5
**Confidence:** 3

**Summary:**

This paper proposes SATA, a motion autoencoder that encodes motion from characters with different skeletal topologies into a unified motion latent space. The key components include semantic-aware feature modulation which leverage LLM-generated joint semantic labels to align joints by their function, and a saptio-temporal interleaved architecture for capturing long-range motion semantics. The paper also include a data pipeline which can unify motion representation from different sources. Experiments on HumanML3D and AniMo4D demonstrate high-fidelity reconstruction, text-driven generation, and zero-shot cross-species motion retargeting.

**Compliance With Llm Reviewing Policy:**

Affirmed.

**Final Justification:**

My concerns have been resolved, and I hence adjust my rating.

**Key Questions For Authors:**

My primary concern is the lack of any analysis on the reliability of the LLM-generated labels. The paper does not even disclose which specific LLM was used.

Could the authors
- provide concrete examples of the LLM's input-output pairs for representative skeletons
- compare the labels produced by different LLMs and report the resulting impact on downstream performance
- demonstrate how the pipeline performs on skeletons with a large number of joints or highly unusual topologies where accurate semantic grounding becomes significantly more challenging?

**Limitations:**

Basically achieved. The authors discuss the limitations of species coverage and the dependence of semantic tag generation on LLM.

**Strengths And Weaknesses:**

**Strengths**

- Address a challenge in unifying motion representation across heterogeneous skeletal topologies.

- The idea of using LLMs to generate semantic joint labels and modulate features by semantic embedding is creative and provides an pairing-free solution for cross-topology correspondence.

- The framework is comprehensive, covering the full pipeline from topology-adaptive BVH processing, semantic-modulated encoding, to diverse downstream tasks.

- Zero-shot cross-species retargeting (e.g., biped to quadruped) is an impressive demonstration of the shared latent space's effectiveness.

**Weaknesses**
- The framework heavily depends on LLM-generated semantic labels, yet no analysis is provided on label reliability and performance on rare skeletons.

- Experimental species diversity is limited to humans and quadrupeds, which falls short of the claimed "universal" goal. Cross-species retargeting lacks quantitative metrics and relies solely on qualitative visualization.

- The use of "emergent" to describe zero-shot retargeting is overclaimed. Since the model explicitly learns a shared semantic latent space, this capability is more "by design" than truly emergent.

- Scalability claims lack supporting evidence: no analysis of performance trends as topology diversity grows.

---

> ### Author Rebuttal · Authors · 2026-03-30
>
> Thank you for your constructive feedback and for highlighting the comprehensiveness of our topology-agnostic framework and its pairing-free correspondence. Due to openreview limits, the visual results we refer to are in this [anonymous repo](https://anonymous.4open.science/r/icml26_3391).
>
> **1. Reliability of LLM Labels, Disclosure, and Input-Output Pairs**
>
> We apologize for the omission. We utilized Gemini 2.5 Pro as our primary LLM to parse the raw joint hierarchies. To ensure reliability, we do not allow LLMs to generate free-form text indiscriminately. As detailed in our prompt design (Appendix Figure 6), LLMs are strictly guided by predefined rules (e.g., Taxonomic Neutrality) and an `[Existing Reference Vocabulary]` when introducing new skeletons.
>
> Here are concrete input-output examples demonstrating how disparate raw names are unified into the same semantic grounding. By mapping distinct anatomical terms to shared functional descriptors, LLMs ensure stable semantic embeddings ($X_t$) across species. We will include these explicit pairings in the revised appendix.
>
> Here is a brief list of joint descriptions for AT-HumanML3D, please refer to *Figure A in repo* for full list.
> ```
> {
> "Hips": "the pelvis and hip root anchor",
> "LowerBack": "a lower back bone segment",
> "Spine": "a spine bone segment",
> "Spine1": "a spine bone segment",
> ...
> }
> ```
>
> **2. Impact of Different LLMs on Downstream Performance**
>
> We acknowledge that unconstrained LLMs might produce varied descriptions for the identical anatomical joint. Therefore, our framework inherently mitigates this issue in our prompt template (Appendix Figure 6), we enforce strict Alignment Constraints during label generation. Specifically, when adding new joint names, the prompt explicitly forces the LLM to select from or conform to an `[Existing Reference Vocabulary]` to avoid ambiguity in text description.
>
> To further quantify the impact of initialization of textual description across different models, we conducted an additional ablation study on gpt-5.4-pro, llama-3.3-70B-instruct, and qwen3-next-80b-a3b-instruct. Due to the limited time during the rebuttal, we implemented our framework on a 10% subset of the AT-HumanML3D training data. In this experiment, we completely replaced the original semantic labels with new ones generated by a different large language model. The empirical results demonstrate that our pipeline is robust with different LLM-generated texts, where the downstream reconstruction performance remained nearly identical. For full table, please refer to *Table A & B in the repo*.
>
> We will include this detailed analysis in the revised appendix.
>
> |LLM|JR↓|RT↓|JP↓|FS↓|GP↓|Internal↓|Cross↓|
> |-|-|-|-|-|-|-|-|
> |Gemini 2.5 Pro|0.1555|1.8504|3.0299|0.0004|0.0004|1.5744|1.4198|
> |Llama 3.3 70B|0.1559|2.0458|3.1362|0.0013|0.0005|1.7486|1.5479|
> |Qwen3-Next 80B|0.1666|2.2954|3.3013|0.0007|0.0005|1.7207|1.6201|
> |GPT-5.4-Pro|0.1652|1.9429|3.0642|0.0011|0.0005|1.5782|1.4661|
>
> **3. Performance on Unusual Topologies and Large Joint Counts**
>
> We evaluated an extreme OOD skeleton: a Snake. Illustrated by *Figure B in the repo*, our model maintained structural mapping. LLM parsed the dense joints using our predefined rules in prompts (Numerical Invariance and Existing Reference Vocabulary). We do observe a drop in physical naturalness. Without specialized motion priors (e.g., slithering) in training data, the output remains structurally aligned but physically limited. These qualitative examples will be added to the revised Appendix.
>
> **4. Terminology: "Emergent" and "Universal"**
>
> We agree with your assessment regarding our terminology. Our original intent in using the word "emergent" was simply to emphasize that the model achieved zero-shot cross-species retargeting without any explicit cross-skeleton pairing or retargeting-specific objective during training.
>
> In the final version, we will replace "emergent" with more accurate phrasing, such as "zero-shot generalization." Furthermore, we will tone down the claim of being "universal" to clarify that the framework is currently "broadly applicable across diverse vertebrates," accurately reflecting the scope of our datasets. We appreciate you pointing this out, as it helps improve the rigor of our writing.
>
>
> **5. Scalability Claims**
>
> To clarify the scalability analysis, Table 5 explicitly evaluates data scaling by progressively adding more species (and thus unique topologies) to the training set. It shows that as structural diversity grows, performance on a fixed, held-out test set consistently improves. Additionally, we scaled our model size using a fixed 10% training subset, please refer to *Table E in repo*.

---

> > ### Author Rebuttal · Reviewer_hWbJ · 2026-04-01
> >
> > I thank the authors for their thorough and well-organized rebuttal. My main concerns have been adequately addressed. I encourage the authors to incorporate the promised additions into the final version.

---

### Official Review · Reviewer_EAZG · 2026-03-11

**Soundness:** 2
**Presentation:** 3
**Significance:** 3
**Originality:** 3
**Overall Recommendation:** 5
**Confidence:** 4

**Summary:**

This paper proposes a Semantic-Aware Topology-Agnostic Motion Autoencoder designed to learn a unified motion representation across heterogeneous skeletal structures. The authors attempt to address a major problem in motion representation learning: the lack of a scalable representation that generalizes across characters with different skeletal topologies and even across species. The authors study the concept of topology-agnostic motion representation by introducing a semantic-aware modulation mechanism and a spatio-temporal interleaved graph architecture that encodes motion into a unified latent manifold and decodes it for arbitrary target skeletons. The framework aims to enable cross-topology reconstruction, retargeting, and downstream generative tasks such as text-to-motion.

**Compliance With Llm Reviewing Policy:**

Affirmed.

**Final Justification:**

I appreciate the authors’ comments, which addressed my concerns. I will maintain my positive score.

**Key Questions For Authors:**

(1) What is the concrete contribution of the semantic-aware modulation beyond structural information?

The proposed method relies heavily on semantic joint descriptions generated by large multimodal models to align heterogeneous skeletons. However, it is not clear from the current experiments whether the performance gains truly originate from semantic grounding or simply from additional conditioning signals. Could the authors provide experiments where semantic labels are randomized, shuffled across joints, or replaced with simple learned embeddings without textual semantics? If the method still performs similarly under such conditions, it would suggest that the semantic component may not be the key factor enabling topology-agnostic learning. Clarifying this would significantly affect my assessment of the novelty of the approach.

(2) How stable and reproducible are the semantic joint descriptions generated by LMMs?

The paper states that semantic descriptions are generated automatically and then encoded as features for each joint. However, the process of generating these descriptions is not described in sufficient detail. Are the semantic tags fixed and manually curated, or dynamically generated by an LMM during preprocessing? If an LMM is involved, how sensitive is the representation to variations in prompt design or language output? Since the semantic features are central to the proposed framework, understanding their reproducibility and robustness is important for evaluating the reliability of the method.

(3) How well does the proposed representation generalize to more structurally diverse articulated systems?

The experiments focus primarily on human skeleton variants and quadrupedal animals. While these represent meaningful variations, they still share certain structural similarities. Have the authors evaluated the method on skeletons with more extreme structural differences, such as robotic manipulators or articulated systems with substantially different joint connectivity patterns? If such experiments are available, they would provide stronger evidence that the learned latent space truly decouples motion semantics from skeletal topology.

(4) How sensitive is the performance to the architectural complexity of the model?

The proposed architecture integrates multiple components, including semantic-aware modulation, graph neural networks, spatial transformers, temporal transformers, and VAE/RVQ-VAE latent regularization. While ablation experiments remove some modules, they do not fully clarify the relative importance of these design choices. Could the authors provide further analysis on whether a simplified architecture (for example, without the transformer components or without the semantic modulation) can achieve comparable performance? Understanding the necessity of each module would help clarify whether the improvements stem from a specific conceptual contribution or from the cumulative effect of multiple architectural techniques.

**Limitations:**

The paper should more clearly articulate the technical limitations of the proposed representation, particularly regarding the reliability of semantic alignment across highly dissimilar skeletons and the potential failure modes when transferring motions between anatomically incompatible species.

**Strengths And Weaknesses:**

Strengths

The work focuses on an important and underexplored problem in motion representation learning. Most existing methods rely on canonical skeleton templates, which severely limits their applicability to heterogeneous datasets or cross-species scenarios. Addressing topology variation is therefore a meaningful direction. The proposed framework is architecturally coherent and integrates several reasonable components. The paper also attempts to validate the approach on both human and animal motion datasets, which is a step toward more general motion representations beyond human-only benchmarks.

Weaknesses

(1) Despite the interesting problem formulation, the central innovation—semantic-aware modulation—remains insufficiently justified both theoretically and empirically. The method relies on semantic joint descriptions generated using large multimodal models, yet the paper does not clearly describe how these semantic tags are constructed, whether they are stable or reproducible, or how much they actually contribute to cross-species alignment. The experimental section does not include a rigorous evaluation of the semantic embeddings themselves, leaving open the possibility that the semantic component functions merely as an auxiliary conditioning signal rather than a fundamental mechanism enabling topology-agnostic learning.

(2) The experimental comparison is also limited and does not convincingly establish the method’s position relative to current state-of-the-art approaches. The evaluation primarily compares against SAME and a few adapted baselines, while many recent generative or representation learning methods for motion modeling are not considered, such as M-R2ET and SAN for motion retargeting, and TapMo and OmniMotionGPT for motion generation. In several cases, topology-dependent baselines are adapted through padding strategies that may disadvantage them relative to the proposed architecture. As a result, the reported improvements may partly reflect differences in experimental setup rather than the intrinsic superiority of the proposed method.

(3) The authors argue that their model learns a unified motion manifold that decouples motion semantics from skeletal topology. However, the experimental validation is limited to human skeleton variants and quadrupedal animals. This range of structures is relatively narrow and does not provide strong evidence that the representation truly generalizes to arbitrary topologies. Without experiments on more diverse or structurally different articulated systems, the claim of a universal topology-agnostic representation appears overstated.

(4) The architecture itself appears somewhat over-engineered. The model combines multiple sophisticated components—including semantic modulation, graph neural networks, spatial transformers, temporal transformers, and latent VAE/RVQ mechanisms—yet the ablation studies do not fully clarify the contribution of each design choice.

[M-R2ET] Zhang, J., Tu, Z., Weng, J., Yuan, J., & Du, B. (2024). A modular neural motion retargeting system decoupling skeleton and shape perception. IEEE Transactions on Pattern Analysis and Machine Intelligence, 46(10), 6889-6904.

[SAN] Aberman, K., Li, P., Lischinski, D., Sorkine-Hornung, O., Cohen-Or, D., & Chen, B. (2020). Skeleton-aware networks for deep motion retargeting. ACM Transactions on Graphics (ToG), 39(4), 62-1.

[TapMo] Zhang, J., Huang, S., Tu, Z., Chen, X., Zhan, X., Yu, G., & Shan, Y. (2023). Tapmo: Shape-aware motion generation of skeleton-free characters. arXiv preprint arXiv:2310.12678.

[OmniMotionGPT] Yang, Z., Zhou, M., Shan, M., Wen, B., Xuan, Z., Hill, M., ... & Wang, Y. (2024). Omnimotiongpt: Animal motion generation with limited data. In Proceedings of the IEEE/CVF Conference on Computer Vision and Pattern Recognition (pp. 1249-1259).

---

> ### Author Rebuttal · Authors · 2026-03-30
>
> We appreciate your constructive suggestions and your recognition of our work as a meaningful step toward general, cross-species motion representation. Due to openreview limits, the visual results we refer to are in this [anonymous repo](https://anonymous.4open.science/r/icml26_3391).
>
> **1. Concrete Contribution of Semantic Modulation**
>
> Following your insightful suggestion, we conducted an inference-time ablation with the model trained on AT-HumanML3D on semantic features, as illustrated in the table below.
>
> - Shuffled Semantics: randomly shuffle the semantic labels across joints.
> - Purely Random Noise: replace semantic tags with Gaussian noise.
>
> While replacing textual semantics with simple learnable embeddings is a standard and logical ablation in fixed-topology tasks, we respectfully highlight that it is intrinsically incompatible with our zero-shot, cross-species setting due to the presence of unseen joints.
>
> |Semantic Cond.|JR↓|RT↓|JP↓|FS↓|GP↓|Internal↓|Cross↓|
> |-|-|-|-|-|-|-|-|
> |Base(Ours)|0.0769|1.1426|1.6311|0.0005|0.0004|1.1986|1.2112|
> |Shuffle|0.5002|33.926|56.001|6.2133|3.1172|88.136|85.950|
> |Random|0.4971|49.3146|68.343|6.6067|3.8787|160.87|118.92|
>
> This explicitly confirms that the semantic component is **not** an auxiliary signal, but a core mechanism that enables topology-agnostic alignment.
>
>
> **2. Stability and Reproducibility of LMM Descriptions**
>
> We apologize for not making this clearer in the main text. The LMM is not used dynamically during runtime. As detailed in Appendix C.3 and Figure 6, the semantic tags are generated via an offline preprocessing step, making our pipeline stable and reproducible.
>
> This offline process creates a fixed Universal Motion Vocabulary (a curated JSON dictionary). During training and inference, the model performs a deterministic lookup from this fixed dictionary. For zero-shot samples during inference, our prompt also instructs the LMM to generate exactly the same text descriptions for the shared joints to reduce ambiguity.
>
> Our model is compatible with semantic tags generated by different LLMs, please refer to response 2 to Reviewer hWbJ for the results. We will release the code containing this vocabulary and the prompt scripts to ensure complete reproducibility.
>
> **3. Generalization Limits**
>
> We appreciate this insightful point. While our current validation focuses on biological entities, the structural diversity within this scope is substantial. Our AT-AniMo4D dataset covers 115 distinct species (e.g., bipeds, quadrupeds, primates, and reptiles), with varied connectivity matrices. Furthermore, we additionally tested our model on snake-like skeletons, where our model preserves most core kinematics. Illustrated in *Video B in the repo*.
>
> We acknowledge that systems like robotic manipulators differ vastly in their kinematic constraints (e.g., fixed-base serial chains and some strictly orthogonal 1-DoF mechanical joints) compared to current biological locomotion. To be precise, we will clarify our "universal" claim in the revision, explicitly stating that our model provides a topology-agnostic representation "across diverse biological articulated systems."
>
>
> **4. Architectural Complexity**
>
> Our architecture integrates multiple components, each of which targets a distinct, non-overlapping necessity to balance local physical constraints and global kinetic coherence. This is validated by a comprehensive ablation study in *Table C in the repo*.
>
> Design choices:
> - GNN (Local MPNN): Strictly enforces local biomechanical constraints.
> - Spatial Transformer: Mitigates GNN over-smoothing by explicitly capturing long-range, non-connected joint synergies.
> - Temporal Transformer (vs. Mean Pooling): Naive mean pooling acts as a low-pass filter, flattening high-frequency kinematic details. The temporal self-attention dynamically preserves complex motion rhythms and phase alignments, resulting in higher motion consistency and quality.
>
> **5. Comparison with Baselines**
>
> We appreciate the references to M-R2ET, SAN, TapMo, and OmniMotionGPT, and we will include a detailed discussion of these recent works in our revised related work section. We focused our comparison on SAME and MoMask/AniMo because our goal is to build a unified, generative autoencoder that handles arbitrary and unaligned topologies in a single forward pass.
>
> However, OmniMotionGPT is strictly tied to fixed SMAL/SMAL-like template parameterizations. M-R2ET and TapMo are highly specialized retargeting/generation frameworks that require predefined pairings or specific shape priors.
>
> Nevertheless, to explicitly address your concern and provide a broader perspective, we adapt and train SAN on AT-HumanML3D as a baseline for retargeting frameworks. For full table, please refer to *Table D in the repo*.
>
> Method|Internal↓|Cross↓|
> -|-|-|
> SAN|15.965|34.821|
> Ours|0.2122|0.1974|

---

> > ### Author Rebuttal · Reviewer_EAZG · 2026-04-01
> >
> > Thank the authors for the rebuttal. My main concerns have been fully addressed.

---

### Official Review · Reviewer_ccCW · 2026-03-12

**Soundness:** 3
**Presentation:** 3
**Significance:** 3
**Originality:** 3
**Overall Recommendation:** 4
**Confidence:** 3

**Summary:**

This paper proposes an improved topology-agnostic motion autoencoder designed to learn a unified latent representation across heterogeneous skeletal structures. The core idea is to use natural language-based skeleton structure semantics to decouple topology from semantic in motion representation learning. In addition, the authors introduce a spatio-temporal interleaved architecture in both the encoder and decoder. This design enables explicit temporal modeling, in contrast to SAME, which operates primarily on a per-frame basis. A graph transformer component is further employed to improve representation capacity.

The authors construct topology-agnostic versions of HumanML3D and AniMo4D and evaluate the model on motion reconstruction, retargeting, and cross-dataset as well as cross-species zero-shot generalization. The proposed method demonstrates improved performance over prior graph-based approaches such as SAME, particularly in cross-dataset and cross-species settings, suggesting enhanced ability to learn topology-independent motion representations. Additional experiments on transfer learning, text-to-motion generation, and qualitative cross-species retargeting further illustrate the quality and versatility of the learned latent space.

**Compliance With Llm Reviewing Policy:**

Affirmed.

**Final Justification:**

After considering the rebuttal, my main concerns regarding the presentation have been sufficiently addressed. I therefore increase the presentation score from 2 to 3, and raise my overall recommendation from 3 (Weak Reject) to 4 (Weak Accept).

**Key Questions For Authors:**

1. **Could you explain the clarity issues listed in the weaknesses?**

2. **Could you provide the total parameter count and hidden dimensions of the proposed model, as well as those of the SAME baseline (as implemented in this paper)?**

3. **Since semantic joint descriptions are central to the method, could the authors provide example full-skeleton joint descriptions used during training?**
   Additionally, have you analyzed how changes in the generated text descriptions (e.g., paraphrasing, simplification) affect performance?

4. **In the zero-shot cross-species retargeting setting, are there cases where the proposed model performs comparably to or worse than the SAME baseline?**
   Are there any ideas for more systematic quantitative analysis of this “emergent ability”? This could also suggest promising future directions.

---

Addressing **Questions 1 and 2** will likely improve the rating.
**Questions 3 and 4** are more open-ended; while responding to them would certainly strengthen the work, I understand that they may fall beyond the scope of this study and could be positioned as directions for future research.

**Limitations:**

Yes.

**Strengths And Weaknesses:**

## Strengths

- This paper tackles an important and challenging problem of learning **topology-irrelevant motion representations** across heterogeneous skeletal structures, including cross-species settings.

- The overall architecture is coherent and carefully designed. The semantic conditioning and the integration of **spatio-temporal graph modeling** are technically reasonable.

- The paper includes a broad range of experiments to validate the proposed method, and consistent performance improvements over prior baselines are observed, demonstrating the practical value of the approach.

- The framework enables **joint training across different motion datasets**, which may have impacts on developing more universal motion representation models.


## Weaknesses

- **Lack of clarity in key architectural details**
  - It is not explicitly described how the **decoder predicts motion features from node representations**. While it is likely implemented as a linear projection or MLP applied to graph node features, this should be clearly specified.
  - **Figure 2** shows the motion feature $F_m$ being passed to a Sinusoidal Embedder, whereas **Equation (1)** applies the Sinusoidal Encoder $\phi_s$ to $[\mathbf{X}_g; \mathbf{X}_l]$. This mismatch creates confusion about the actual pipeline.
  - The paper states that **Large Multimodal Models (LMMs)** are used to generate joint-level semantic descriptions, but it does not clearly specify which model is used. It is also not clear which model is used to embed text into skeleton semantic features (although one can infer from citation that **T5** is used). Since semantic conditioning is central to the contribution, the exact pipeline for **text generation and embedding** should be clearly and explicitly introduced rather than implicitly inferred.
  - **Figure 2** mentions a latent space $z_m$, but only $z_m^{\prime}$ appears in the figure. It is also unclear whether the encoder and decoder share the same architecture; this can only be inferred from the word *“symmetrical”* in the caption. More explicit clarification would improve readability.
  - **Line 166 (left):** static skeleton features $F_s$ are defined through $X_g, X_l \in \mathbb{R}^3$, which appear to be joint-level features rather than global skeleton features. Later, $F_m \in \mathbb{R}^{J \times D}$ is introduced without defining $J$ and $D$. Such notation gaps make the formulation harder to follow.
  - In the **“Temporal Modeling Branch”** paragraph, $\mathcal{G}\_{batch}$ and $\mathcal{X}_{temp}^{\prime}$ are introduced without sufficient explanation.
  - Two options are provided for **latent modeling (VAE and RVQ-VAE)**, but the paper does not provide clear intuition or justification for introducing both instead of focusing on one. It is also unclear which variant is used in experiments in **Tables 1 and 2**.
  - It is unclear which **model configuration** is used in **Figure 4** (e.g., which dataset it is trained on). This information appears only in supplementary videos and should be explicitly stated in the paper.
  - It is unclear in the main text how to extend the model to **text-to-motion generation**. While the appendix provides more details, the core mechanism should be briefly described in the main paper for clarity.
  - If **overlapping sliding windows** are used, the paper does not describe how predictions in overlapping regions are merged (e.g., averaging, cropping, or weighted blending). This detail is important for reproducibility.

- **Missing model capacity details**
  - The paper does not report **model parameters or hidden dimensions of the GNN/transformer blocks**. Without this information, it is difficult to assess whether improvements come from architectural innovation or substantially larger model capacity compared to baselines.

- **Limited analysis of semantic embeddings**
  - Since **semantic embeddings** are central to the method, it would strengthen the paper to analyze how performance changes under **variations or perturbations of the text descriptions**.

- **Limited evidence for cross-species generalization**
  - According to the supplementary materials, the baseline **SAME** model also exhibits some cross-species retargeting ability.
  - While the proposed method demonstrates improved qualitative motion quality in selected examples, the claim of **emergent cross-species capability** would benefit from more systematic and controlled evaluation (e.g., scaling analysis with data size, model capacity, and compute).

---

> ### Author Rebuttal · Authors · 2026-03-30
>
> We thank the reviewer for their careful reading and recognizing our architecture's potential for universal motion representation. Due to openreview limits, the visual results we refer to are in this [anonymous repo](https://anonymous.4open.science/r/icml26_3391).
>
> **1. Key Architectural Details**
>
> - Decoder Prediction: The final layer of the decoder outputs motion representations $F_m^{out}=(q,r,c)\in \mathbb{R}^{J\times 11}$, which can be used to recover human motion through Forward Kinematics. $J$ is the number of joints.
> - Figure 2 vs. Eq 1 Mismatch: Equation (1) is correct. We will correct the arrow in Figure 2 to accurately reflect this routing.
> - LMM & Text Embedding: We used Gemini 2.5 Pro for generating the semantic descriptions via the prompt in Fig. 6. We used the frozen T5 model to embed these text descriptions into the semantic features $X_t$. We will state this explicitly in Section 3.2.
> - Latent Space & Architecture: We will unify the notation to $z_m$ throughout. Yes, the encoder and decoder share identical configurations.
> - Skeleton Feature: Yes, it should be of shape $\mathbb{R}^{J\times 3}$. We will clarify it in the revised version.
> - Temporal Modeling Branch: $G_{batch}$ represents the batched sequence of graphs, and the mapping $\mathcal{T}$ reshapes the temporal dimension for the Transformer to process sequences of node trajectories $\mathcal{X}_{temp}$. We will add this clarification.
> - Latent Modeling: Tables 1 and 2 report the VAE variant. We introduced both to demonstrate compatibility with different generative backbones: VAE is suitable for continuous latent diffusion, while RVQ-VAE is essential for discrete token-based models.
> - Models for Figure 4: Figure 4 is generated using the jointly trained VAE model on both datasets.
> - Extending to text-to-motion: We will add a concise summary to Section 5.3 explaining how we bridge continuous latents with MDM and discrete tokens with MoMask.
> - Overlapping sliding windows: Overlapping regions are generated by the preceding window, and the overlapping portion in the latter window is cropped.
>
> **2. Model Capacity**
> Our improvements stem from architectural innovation rather than merely increasing model capacity. For a fair comparison, we intentionally scaled the SAME baseline with larger hidden layers to match our setup closely.
>
> |Model|N.Param(M)|GFLOPs|
> |-|-|-|
> |Ours|8.41|29.66|
> |SAME|5.98|20.21|
>
> Both models operate within a lightweight regime. The modest increase in our model is mainly attributed to the Graph Transformer, which is essential for topology-agnostic spatio-temporal modeling.
>
> **3. Joint Descriptions & Changes in Text Descriptions**
>
> **Joint Descriptions.** Due to space limits, please refer to *Figure A in the repo*.
>
> **Changes in Text Descriptions.** We rephrased the descriptions. For example, `the pelvis and hip root anchor`-> `Pelvis` (Variant A) -> `Hips` (Variant B). Please refer to response 3 to Reviewer pgeR for the results.
>
> **4. Quantitative Analysis & Scaling**
>
> **Systematic Quantitative Analysis**: As a preliminary attempt to quantify this emergent zero-shot ability, we explored a Cycle Consistency Evaluation inspired by [A]. By retargeting 100 human motions across 5 animal topologies, we constructed an initial proxy of 500 test pairs to evaluate core semantic preservation. We acknowledge that establishing a fully rigorous cross-species metric involves complex nuances. Therefore, we position the formalization and deeper discussion of this evaluation protocol as a key scope for future work.
>
> Notably, even without being optimized for this task, our model achieves significantly lower cycle consistency errors than SAME.
>
> |Method|Recon:JR↓|Recon:RT↓|Recon:JP↓|Recon:FS↓|Recon:GP↓|
> |-|-|-|-|-|-|
> |SAME|0.4232|33.956|47.522|0.4348|0.3639|
> |Ours|0.1717|3.9671|6.8111|0.0003|0.0001|
>
> [A] Zhu, Jun-Yan, et al. "Unpaired image-to-image translation using cycle-consistent adversarial networks." ICCV, 2017.
>
> **Scaling with data size, model capacity and compute**: Table 5 explicitly evaluates data scaling by progressively adding more species (and thus unique topologies). We also scale the model size with the table below. For full table, please refer to Table E in the repo.
>
> |Model|Params(M)|GFLOPs|JR↓|RT↓|JP↓|FS↓|GP↓|
> |-|-|-|-|-|-|-|-|
> |Base|8.41|29.7|0.6201|5.2581|14.344|0.0579|0.1769|
> |Small|3.60|12.8|0.6879|7.1484|16.979|0.0338|0.0739|
> |Tiny|1.62|5.81|0.6957|7.9837|19.462|0.0385|0.1064|
>
> **5. Failure Cases**
>
> While our model outperforms SAME in most zero-shot cross-species scenarios, both models can fail in some cases. Constrained by the absence of paired cross-species ground truth and explicit biomechanical annotations in standard datasets, the zero-shot inference tends to preserve the source's local rotations without accounting for physical dynamics (see *Video C in repo*). Conversely, SAME completely collapses with severe topological distortion. We will discuss integrating physics-based priors as a future direction.

---

> > ### Author Rebuttal · Reviewer_ccCW · 2026-04-03
> >
> > Thank you for your detailed and thorough rebuttal. The main concerns regarding clarity and model size have been addressed. The preliminary results evaluating emergent zero-shot capabilities are also promising. That said, I agree this remains an interesting question that needs more comprehensive investigation beyond the rebuttal, ideally as part of future work.I will adjust my rating from 3 (Weak Reject) to 4 (Weak Accept).

---

### Official Review · Reviewer_pgeR · 2026-03-13

**Soundness:** 2
**Presentation:** 2
**Significance:** 2
**Originality:** 2
**Overall Recommendation:** 4
**Confidence:** 3

**Summary:**

This paper studies motion representation learning across characters with different skeletal topologies, including humans and animals. It proposes a topology-agnostic motion autoencoder that uses semantic joint information and spatio-temporal modeling to learn a shared latent motion space independent of skeleton structure. The paper also introduces a topology-adaptive data pipeline for training on heterogeneous raw motion data. The main contribution is a unified motion representation that supports reconstruction, retargeting, and zero-shot cross-species transfer within a single framework.

**Compliance With Llm Reviewing Policy:**

Affirmed.

**Final Justification:**

All my concerns have been addressed, I'd love to raise my score.

**Key Questions For Authors:**

See weakness.

**Limitations:**

Yes

**Strengths And Weaknesses:**

Strengths：
1.This paper is well-written, easy to follow.
2.The visualizations in the main paper and supplementary material look satisfactory and support the claims of zero-shot motion retargeting.
3.The proposed semantic-aware motion encoding is novel and effective in current research community.

Weakness:
1.Its framework is somewhat incremental and is essentially a conditional VAE. Many components, such as graph spatial modeling, temporal transformers, and the VAE/RVQ bottleneck, have been widely used in prior work.
2.In Table 1 and Table 2, the proposed method achieves inferior FS/GP compared to SAME, leading to a concern that the improvements in JR/RT/JP may come at the expense of local physical plausibility, especially in terms of foot-ground interaction and motion realism.
3.The joint-level semantic priors are derived from relatively clean datasets such as AT-HumanML3D and AT-AniMo4D. It remains unclear whether the model is robust to noisy, incomplete, or ambiguous semantic priors.
4.All experiments are conducted on two transformed benchmarks, AT-HumanML3D and AT-AniMo4D. As a result, the zero-shot generalization ability on more unconstrained or out-of-distribution characters is not sufficiently validated.

If the authors solved all my concerns, I'd love to raise my rating.

---

> ### Author Rebuttal · Authors · 2026-03-30
>
> We sincerely thank the reviewer for recognizing the novelty of our semantic-aware encoding and the effectiveness of our zero-shot visualizations. We highly value your insightful suggestions. Due to openreview limits, the visual results we refer to are in this [anonymous repo](https://anonymous.4open.science/r/icml26_3391).
>
> **1. Question on design choices.**
>
> We appreciate the reviewer's observation. However, our design is driven by the specific challenges of topology-agnostic motion modeling, rather than simply stacking existing modules on a conditional VAE.
>
> While our key insight and core contribution lie in the semantic injection via Semantic-Aware Feature Modulation to decouple motion from rigid skeletal topologies, handling such complexity still requires strong foundational architectures. Within this architecture, each standard component serves a distinct, indispensable function: GNNs and Temporal Transformers are leveraged to model the intricate spatio-temporal motion dynamics, whereas the VAE (or RVQ) bottleneck optimizes and regularizes the latent features, ensuring their suitability for downstream generation.
>
> We will explicitly clarify this design philosophy and the functional division of labor in the revised manuscript.
>
> **2. Question about FS/GP compared to SAME.**
>
> We appreciate the reviewer's attention to detail. We would like to clarify this observation by analyzing the zero-shot and in-distribution scenarios separately:
>
> **Zero-Shot Setting**: SAME seems to have lower FS and GP scores, but these are actually misleading artifacts of catastrophic structural failure, evidenced by extremely large RT and JP errors. The character often suffers from severe global distortion or is left entirely floating in the air, which will also result in a lower FS/GP. Our method prioritizes global structural fidelity compared with SAME, which contains more plausible actions.
>
> **In-Distribution Setting**: Under normal in-distribution conditions, the performance of our method is highly comparable to SAME. The numerical differences are extremely marginal (e.g., FS: 0.0079 vs. 0.0091; GP: 0.0021 vs. 0.0064). In practice, this slight variance does not translate to noticeable visual artifacts. For visualization, please refer to video 1 and video 2 in the previously submitted supplementary.
>
> **3. Model's robustness to ambiguous semantic priors.**
>
> We would first like to emphasize that raw BVH labels are inherently unstandardized and often contain confusing joint names, such as `def_frontLegLwrAllTwist_joint.L` and `def_rearLegLwrAllTwist_joint.R`. As detailed in Appendix C.3, our pipeline does not rely on manually cleaned labels. Instead, we leverage LLMs to autonomously map these raw, abbreviated, and messy joint inputs into standardized semantic labels, demonstrating robust adaptability to ambiguous initial conditions.
>
> To directly address the specific concern, we remove words and rephrase the description. For example, `the pelvis and hip root anchor`-> `Pelvis` (Variant A) -> `Hips` (Variant B). Results are shown in the table below. Our model can process these descriptions with negligible performance loss after training. Details can be found in *Figure D in the repo*.
>
> |Prompt Template|Recon:JR↓|Recon:RT↓|Recon:JP↓|Recon:FS↓|Recon:GP↓|Retarget:Int.↓|Retarget:Cross↓|
> |-|-|-|-|-|-|-|-|
> |Original|0.1555|1.8504|3.0299|0.0004|0.0004|1.5744|1.4198|
> |Variant A|0.1636|2.5392|3.7029|0.0010|0.0003|1.6766|1.5326|
> |Variant B|0.1573|2.2694|3.3667|0.0004|0.0003|1.5708|1.4288|
>
> **4. Zero-shot on more unconstrained characters**
>
> We would like to gently clarify that the data splitting protocol we employed inherently constitutes an OOD evaluation. As detailed in Appendix C.2, test species were entirely excluded from training, meaning all reported cross-species evaluations are on unseen topologies. To explore our model's absolute boundaries, we further tested extremely divergent wild creatures (a limbless snake), with visual results in *Video B in the repo*. Though it is imperfect due to profound structural gaps, our model captures underlying kinematic rhythmic patterns without collapsing. We will discuss these performance boundaries in the revised version.

---

> > ### Author Rebuttal · Reviewer_pgeR · 2026-04-01
> >
> > Thank authors' responses, my concerns have been addressed.

---

### Decision · Program_Chairs · 2026-04-30

**Decision:**

Accept (regular)

**Comment:**

This work studies motion representation learning across characters with different skeletal topologies, including humans and animals. Most of the weaknesses that were raised by reviewers were addressed by the authors. I thikn the scores were already pretty high, but yes, I think so.